# An operational procedure for rapid flood risk assessment in Europe

**Francesco Dottori, Milan Kalas, Peter Salamon, Alessandra Bianchi, Lorenzo Alfieri, Luc Feyen**

European Commission, Joint Research Centre, Directorate for Space, Security and Migration, Via E. Fermi 2749, 21027 Ispra, Italy.

francesco.dottori@ec.europa.eu

**Keywords:** real-time, early warning system, flood hazard mapping, flood impact, economic damage, population, risk assessment

## *Abstract*

The development of methods for rapid flood mapping and risk assessment is a key step to increase the usefulness of flood early warning systems, and is crucial for effective emergency response and flood impact mitigation. Currently, flood early warning systems rarely include real-time components to assess potential impacts generated by forecasted flood events. To overcome this limitation, this study describes the benchmarking of an operational procedure for rapid flood risk assessment based on predictions issued by the European Flood Awareness System (EFAS). Daily streamflow forecasts produced for major European river networks are translated into event-based flood hazard maps using a large map catalogue derived from high-resolution hydrodynamic simulations. Flood hazard maps are then combined with exposure and vulnerability information, and the impacts of the forecasted flood events are evaluated in terms of flood-prone areas, economic damage and affected population, infrastructures and cities.

An extensive testing of the operational procedure has been carried out by analysing the catastrophic floods of May 2014 in Bosnia-Herzegovina, Croatia and Serbia. The reliability of the flood mapping methodology is tested against satellite-based and report-based flood extent data, while modelled estimates of economic damage and affected population are compared against ground-based estimations. Finally, we evaluate the skill of risk estimates derived from EFAS flood forecasts with different lead-times and combinations of probabilistic forecasts. Results highlight the potential of the real-time operational procedure in helping emergency response and management.

## 1) Introduction

Nowadays, flood early warning systems (EWS) have become key components of flood management strategies for many rivers (Cloke et al., 2013; Alfieri et al., 2014a). Their use can increase preparedness of authorities and population, thus helping to reduce negative impacts (Pappenberger et al., 2015). Early warning is particularly important for cross-border river basins where cooperation between authorities of different countries may require more time in order to inform and coordinate actions (Thielen et al., 2009).

In this context, the European Commission has developed the European Flood Awareness System (EFAS) which provides operational flood predictions in major European rivers as part of the Copernicus Emergency Management Services. The service has been fully operational since 2012 and is available to hydro-meteorological services with responsibility for flood warning, EU civil protection, and their networks.

While EWS are routinely used to predict flood magnitude, there is still a gap in their ability to translate flood forecasts into risk forecasts - in other words, to evaluate the possible consequences generated by forecasted events (e.g. flood-prone areas, affected population, flood damages and losses), given their probability of occurrence. Generally, flood impacts are evaluated considering reference risk scenarios where a fixed return period is used for all of the area of interest, for instance based on official maps issued by competent authorities (EC, 2007). However, this implies some degree of interpretation to define flood impact and risk in case of a flood forecast. Some research projects are being developed where flood impact estimation is automated and linked to event forecasting (Rossi et al., 2015; Schulz et al., 2015; Saint-Martin et al., 2016). However to our knowledge these systems are still at an experimental phase, and are not yet integrated into operational EWS.

The availability of real-time operational systems for assessing potential consequences of forecasted events would be a substantial advance in helping emergency response (Molinari et al., 2013), and indeed flood risk forecasts are increasingly being requested by end-users of early warning systems (Emerton et al., 2016; Ward et al., 2015). At a local scale, the joint evaluation of flood probabilities and consequences may not only increase preparedness of emergency services, but also allow cost-benefit considerations for planning and prioritizing response measures (e.g. strengthening flood defences, planning evacuation of people at risk). At a European scale, the possibility to receive prior information on expected flood risk would help the Emergency Response Coordination Centre (ERCC) in prioritizing and coordinating support to national emergency services.

In the present paper, we describe a methodology that is designed to meet the needs of EWS users and to overcome the limitations mentioned so far. The methodology translates EFAS flood forecasts into event-based flood hazard maps, and combines hazard, exposure and vulnerability information to produce risk estimations in near real-time. All the components are fully integrated within the EFAS forecasting system, thus providing seamless risk forecasts at European scale.

To demonstrate the reliability of the proposed methodology, we perform a detailed assessment
focused on the 2014 floods in the Sava River Basin in Southeast Europe. A large dataset for the
evaluation of the results has been collected, consisting of observed flood magnitude, flood extent
derived from different satellite imagery datasets, and detailed post-event evaluation of flood
impacts, economic damage assessment and affected population and infrastructure.
The reliability of the flood mapping procedure is first assessed by assuming a "perfect" forecast,
where flood magnitude is taken from real observations instead of EFAS predictions. The effect
of the failure of flood defences is also taken into account. Subsequently, we test the performance
of the operational flood forecasting procedure, to evaluate the influence of different lead-times
and combinations of forecast members.

## 85 *2) Methodology*


In this Section we describe the three components which comprise the rapid risk assessment
procedure: 1) streamflow and flood forecasting; 2) event-based rapid flood hazard mapping 3)
impact assessment. Figure 1 shows a conceptual scheme of the steps comprising the methodology.

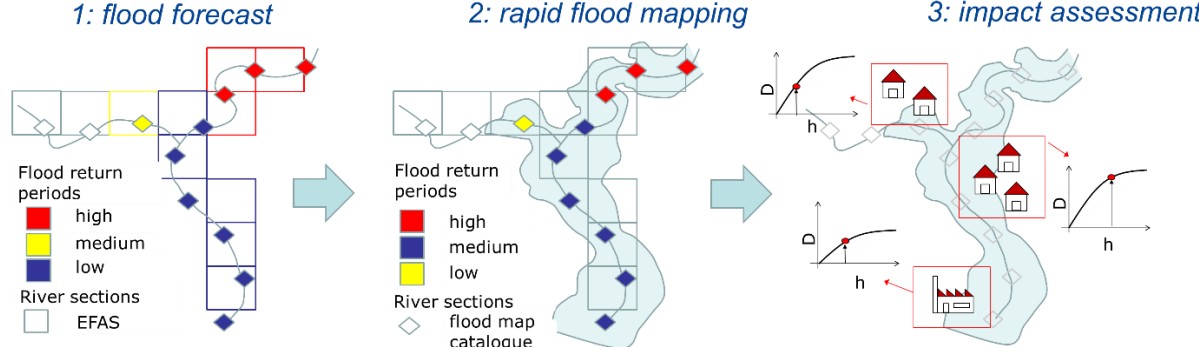

*Figure 1: Conceptual scheme of the rapid risk assessment procedure*

The basic workflow of the procedure is outlined as follows:
• Every time a new forecast is available, the procedure defines the river sections potentially
affected and local flood magnitude, expressed as the return period of the peak discharge;
• Areas at risk of flooding are identified using a map catalogue, which defines all the flood-
prone areas for each river section and flood magnitude; these local flood maps are then
compared against local flood protection levels and merged to derive event-based hazard maps;
• Event hazard maps are combined with exposure and vulnerability information to assess
affected population, infrastructures and urban areas, and economic damage.

The described procedure is fully integrated within the existing EFAS forecast analysis chain and
operates in near real-time. When a new EFAS hydrological forecast becomes available (Step 1),
the risk assessment procedure is activated for those locations where predicted peak discharges
exceed the flood protection levels (Step 2). When activated, the execution time depends on the
extent and spatial spread of the affected areas over the full forecasting domain. Even in the case
of flood events occurring simultaneously in different European countries, the results of the
analysis are delivered within one hour after the EFAS forecast runs are finished.
The following Sections provide a detailed description of each component.

### 111 *2.1 Flood forecast: the European Flood Awareness System (EFAS)*


The European Flood Awareness System (EFAS) produces streamflow forecasts for Europe using
a hydrological model driven by daily weather forecasts. Below we provide a general description
of the EFAS components. For further details the reader is referred to the EFAS web-site
(www.efas.eu) and to published literature (Thielen et al., 2009; Pappenberger et al., 2011; Cloke
et al., 2013; Alfieri et al., 2014a).
Hydrological simulations in EFAS are performed based on LISFLOOD (Burek et al, 2013; van
der Knijff et al., 2010), a distributed physically-based rainfall-runoff model combined with a
routing module for river channels. The model is calibrated at European scale using streamflow
data from a large number of river gauges, and meteorological fields interpolated from point
measurements of precipitation and temperature. Based on this calibration, a reference
hydrological simulation for the period 1990-2013 is run for the European window at 5 km grid
spacing, and updated daily. This reference simulation provides initial conditions for daily forecast
runs of the LISFLOOD model driven by the latest weather predictions, which are provided twice
per day with lead-times up to 10 days. The reference simulation is also used to estimate discharge
values for the return periods corresponding to 1, 2, 5 and 20 years, at every point of the river
network. All flood forecasts are compared against these discharge thresholds and the threshold
exceedance is calculated. If the 5-year threshold is consistently exceeded over three consecutive
forecasts, flood warnings for the affected locations are issued to the members of the EFAS
consortium. The persistence criterion has been introduced to reduce the number of false alarms
and to focus on large fluvial floods caused mainly by widespread severe precipitation, combined
rainfall with snow-melting, or prolonged rainfalls of medium intensity.
To account for the inherent uncertainty of the weather forecasts, EFAS adopts a multi-model
ensemble approach, running the hydrological model with forecasts provided by the European
Centre for Medium Weather Forecast (ECMWF), the Consortium for Small-scale Modelling
(COSMO), and the Deutscher Wetterdienst (DWD).

### 138 *2.2 Rapid flood hazard mapping*

### 139 *2.2.1 Database of flood hazard maps*


Linking streamflow forecast with inundation mapping is complex because inundation modelling
tools are computationally much more demanding than hydrological models used in EWS, which
currently prevent a real-time integration of these two components. To overcome this limitation,
in this study we have created a catalogue of flood inundation maps, covering all of the EFAS river
network and linked to EFAS streamflow forecasts.
The hydrological input for creating the map catalogue is derived from the streamflow dataset of
the EFAS reference simulation, described in Section 2.1. The information is available for the
EFAS river network at 5 km grid spacing for rivers with upstream drainage areas larger than 500
km$^2$. Since hydrographs simulated in the EFAS reference simulation do not refer to specific return
periods, we use a statistical analysis of extreme values to derive peak discharges for every cell of
the river network for reference return periods of 10, 20, 50, 100, 200 and 500 years. In addition,
we extract flow duration curves from the reference simulation, which are used together with peak
discharges to calculate synthetic flood hydrographs (see Alfieri et al., 2014b for a detailed
description).
The streamflow data are then downscaled to a high-resolution river network (100 m), where
reference sections are identified at regular spacing along the stream-wise direction every 5 km.
100 m sections are then linked to a section of the 0.1° river network, in order to assign to each
section a synthetic discharge hydrograph. Where the coarse- and high-resolution river networks
do not overlap, flood points are linked with the closest 0.1° pixel in the upstream direction. Note
that there is not a one-to-one correspondence between 5 km and 100 m river sections. In particular,
some 5 km sections have no related sections in the 100 m river network, while others can have
more than one. Figure 2 shows a conceptual scheme of the two river networks. The digital
elevation model (DEM) used to derive the 100 m river network is a component of the River and
Catchment Database developed at JRC (Vogt et al., 2007). The same DEM is used also to run
flood simulations at 100 m resolution for each 100 m river-section using the 2D hydrodynamic
model LISFLOOD-FP (Bates et al., 2010), fed with synthetic hydrographs. Therefore, for every
100 m river section we derive flood maps for the six reference return periods.
The flood maps related to the same EFAS river section (i.e. pixel of the 5 km river network) are
merged together, to identify the areas at risk of flooding due to overflowing from a specific EFAS
river section, and archived in the flood map catalogue. The merging is performed separately for
each return period, in order to relate flooded areas with the magnitude of the flood event.

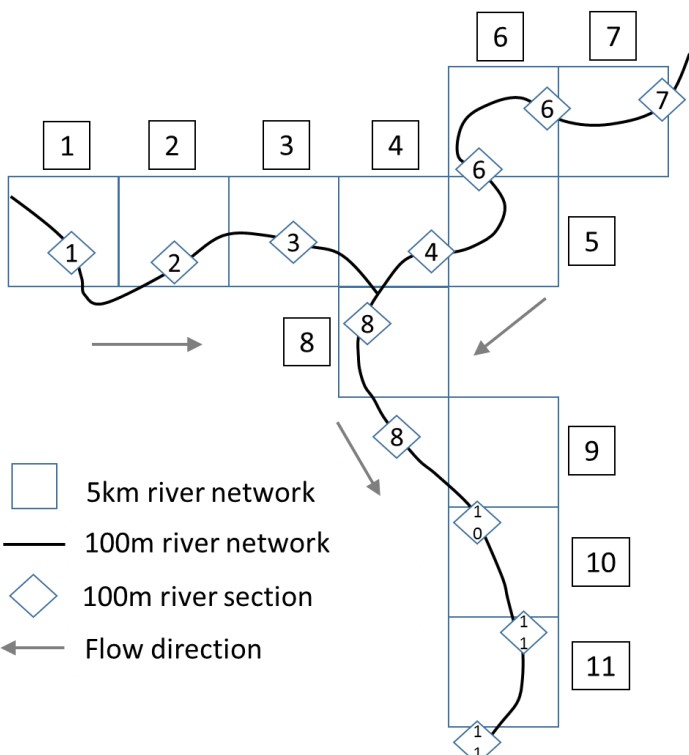

*Figure 2: Conceptual scheme of the EFAS river network (5 km, squares) with the high-resolution network (100 m) and river sections (diamonds) where flood simulations are derived. The related sections of the two networks are indicated by the same number. Adapted from Dottori et al. (2015).*

### 2.2.2 Event-based mapping of flood hazard

This step of the procedure provides a rapid estimation of the expected flood hazard, using the database of flood maps described in Section 2.2.1 to translate EFAS discharge forecasts into event-based flood mapping.

At each grid cell, we first identify the median of the ensemble forecast given by the latest EFAS prediction, and then select the maximum discharge of the median over the full forecasting period (10 days). This value is compared with the reference long-term climatology to calculate the return period. In this way, the range of ensemble forecasts is taken as a measure of the probability of occurrence, while forecast return periods allow estimation of the magnitude of predicted flood events. Then, predicted streamflow is compared with the local flood protection level, and river grid cells where the protection level is exceeded are considered to activate the impact assessment procedure. Flood protection levels are given as the return period of the maximum flood event which can be retained by the defence measures (e.g. dykes). The map of flood protections used is based on risk-based estimations for Europe developed by Jongman et al. (2014), integrated (where available) with the actual level of protection found in a literature review or assessed by local

authorities (see Appendix for more details). Note that flood protections are not considered in
LISFLOOD-FP simulations, because at a European scale there is no consistent information about
the location and geometry of flood protection structures (e.g. levees). As such, LISFLOOD-FP
simulations are run as if there were no protection structures.
Selected river cells are reclassified according to the closest return period exceeded (10, 20, 50,
100, 200, 500 years), and the corresponding flood hazard maps are retrieved from the catalogue
and tiled together. For instance, if the estimated return period is 40 years, the flood map for 20
years return period is used. Where more maps related to more river sections overlap (see Section
2.2), the maximum depth value is taken.

## 2.3 Flood impact assessment



After the event-based flood hazard map has been completed, it is combined with the available
information defining the exposure and vulnerability at European scale.
The number of people affected is calculated using the population map developed by Batista e
Silva et al. (2012) at 100 m resolution. A detailed database of infrastructures produced by Marín
Herrera et al. (2015) is used to compute the extension of the road network affected during the
flood event. The list of major towns and cities potentially affected within the region is derived
from the map of World Cities developed by ESRI (2017).The total extension of urban and built-
up areas (differentiated between residential, commercial and industrial areas) and agricultural
areas is computed using the latest update of the Corine Land Cover for the year 2012 (Copernicus
LMS, 2017).
The land use layer also provides the exposure information to compute direct economic losses in
combination with flood hazard variables and flood damage functions, following the approach
developed by Huizinga et al. (2007). More specifically, we use a set of normalized damage
functions to calculate the damage ratio as a function of water depth, ranging from 0 (no damage)
to 1 (maximum damage). The damage ratio is then multiplied by the maximum damage value,
calculated as a function of land use and the country's GDP, to calculate actual damage. Separate
damage functions are applied for the land use classes that are more vulnerable to flooding
(residential, commercial, industrial, agricultural). In addition, to account for variations in value
of assets within a country, damage values are corrected considering the ratio between the gross
domestic product (GDP) of regions (identified according to the Nomenclature of Territorial Units
for Statistics (NUTS), administrative level 1) and the country's GDP.
For countries where specific damage functions could be found in literature, Huizinga et al. (2007)
produced normalized functions based on these national data. In addition, the same authors
elaborated averaged functions to be used for countries without national data, in order to produce
a consistent dataset at European scale. The same approach has been applied in the present study
to elaborate damage curves for countries not included in the original database, such as Serbia and
Bosnia-Herzegovina. The complete set of damage functions and the detailed description of the
methodology, are available as supplementary data of the recent report by Huizinga et al. (2017).
All the results computed during the risk assessment procedure are aggregated using the
classification of EU regions of EUMetNet (the network of European Meteorological Services,
www.meteoalarm.eu). The regions considered are based on Levels 1 and 2 of the NUTS
classification, according to the EU country, with the advantage of providing areas of aggregation
with a comparable extent.

## 238  *3) Benchmarking of the procedure*

In order to perform a comprehensive evaluation of the risk assessment procedure, it is important
to evaluate each component of the methodology, i.e. streamflow forecasts, event-based flood
mapping, and the impact assessment. The skill of EFAS streamflow forecasts is routinely
evaluated (Pappenberger et al., 2011) while impact assessment has been successfully applied by
Alfieri et al. (2016) to evaluate the socio-economic impacts of river floods in Europe for the
period 1990-2013. Here, the complete procedure is tested using the information collected for the
catastrophic floods of May 2014, which affected several countries in Southeast Europe. In
particular, we focus on the flooding of the Sava River in Bosnia-Herzegovina, Croatia and Serbia.

### 248  *3.1 The floods in Southeast Europe in May 2014*

Exceptionally intense rainfalls, from 13 May 2014 onwards, following weeks of wet conditions,
led to disastrous and widespread flooding and landslides in South-Eastern Europe, in particular
Bosnia-Herzegovina and Serbia. In these two countries, the flood events were reported to be the
worst for over 200 years. More than 60 people lost their lives and over a million inhabitants were
estimated to be affected, while estimated damages and losses exceeded 1.1 billion Euro for Serbia
and 2 billion Euro for Bosnia-Herzegovina (ECMWF, 2014; ICPDR and ISRBC, 2015). Critical
flooding was also reported in other countries including Croatia, Romania and Slovakia. Serbia
and Croatia requested and obtained access to the EU Solidarity Fund for major national disasters
(EC, 2016).
According to the Technical Report issued by the International Commission for the Protection of
the Danube River and the International Sava River Basin Commission (ICPDR and ISRBC,
2015), the flood events were particularly severe in the middle-lower course of the Sava River and
in several tributaries. The discharge measurements and estimations carried out between 14-17
May indicated that peak flow magnitude exceeded the 500 years return period both in the Bosna
and Kolubara rivers and in part of the Sava River downstream of the confluence with Bosna.
Discharges above 50 years were observed in the Una, Vrbas, Sana and Drina rivers (Figure 3).

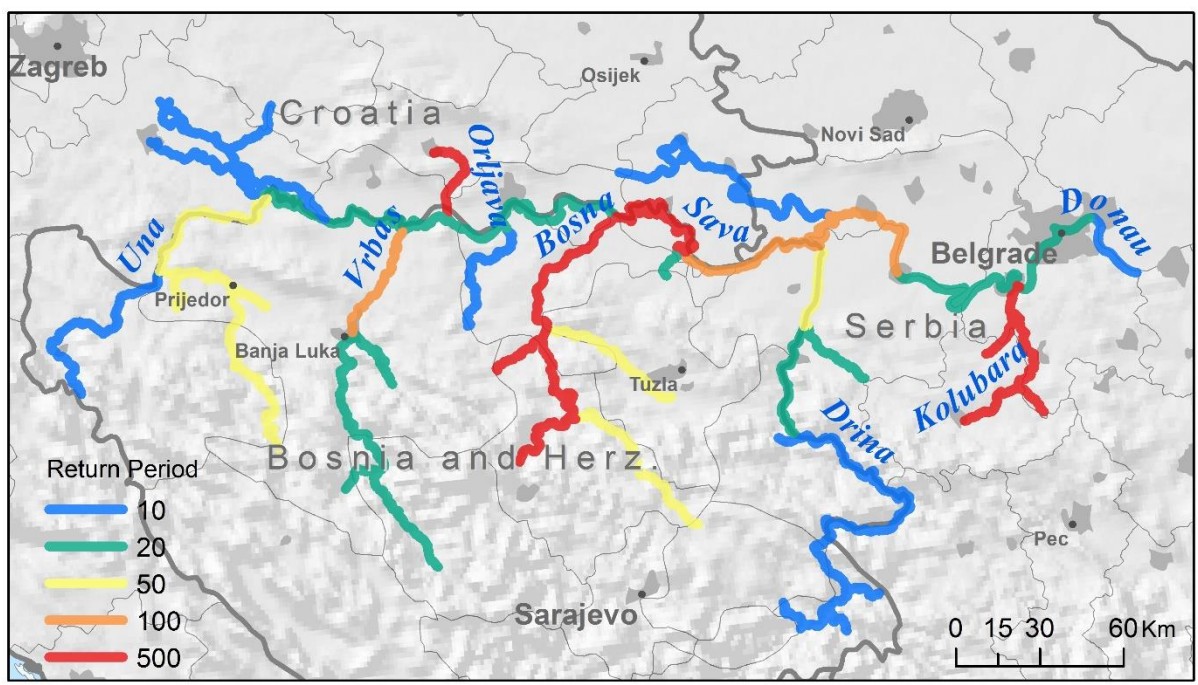

*Figure 3. Reconstruction of return period of peak discharges in Sava River basin (source:*
*ICPDR and ISRBC, 2015).*

The lower reach of the Sava was less heavily affected because upstream flooding reduced peak
discharges, and hydraulic operations on the Danube hydraulic structures reduced water levels in
the Danube (ICPDR and ISRBC, 2015). Due to the extreme discharges, multiple dyke breaches
occurred along the Sava River, and severe flooding occurred at the confluence of tributaries such
as Bosna, Drina and Kolubara (Figure 4). In many areas, dykes were reinforced and heightened
during the flood event to withstand the peak flow; additional temporary flood defences were also
built to prevent further flooding, and drains were dug to drain flooded areas more quickly. Other
rivers in the area experienced severe flood events, such as the tributaries of the Danube Velika
Morava and Mlava, in Serbia.
Table 1 reports a summary of flood impacts at national level for Bosnia-Herzegovina, Croatia and
Serbia, retrieved from different sources.

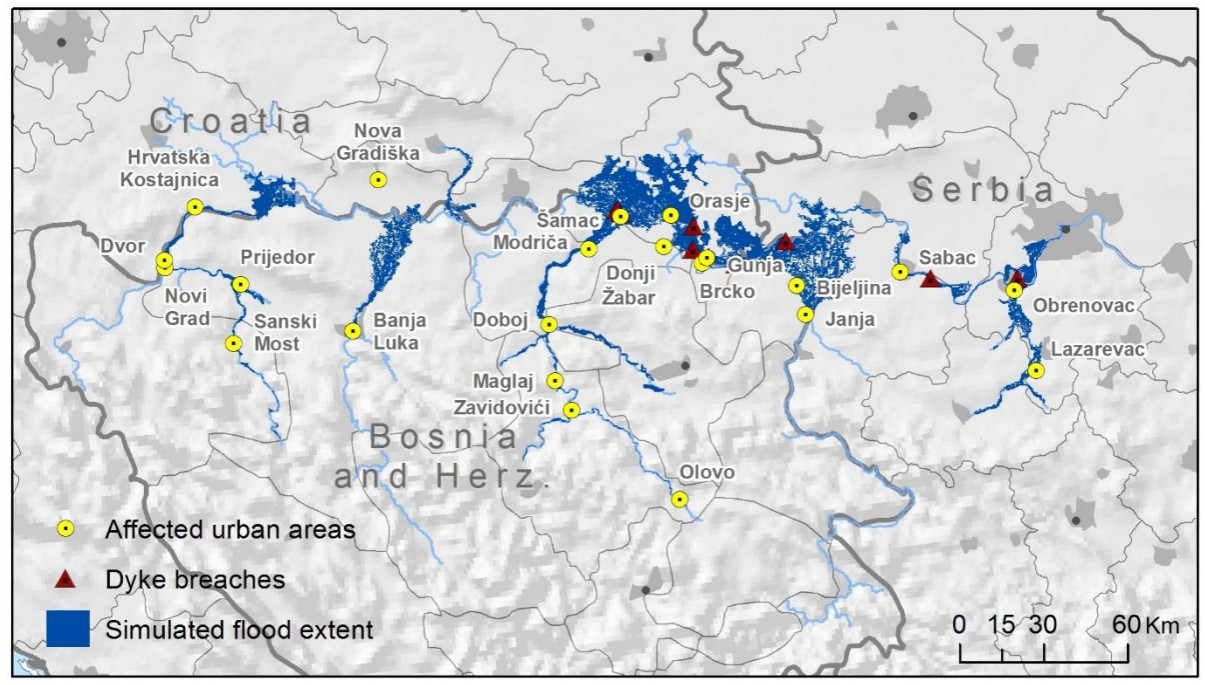

*Figure 4. Reconstruction of affected urban areas and dyke failure locations along the Sava*
*River (sources: UNDAC, 2014; ICPDR and ISRBC, 2015). The flood extent of the reference*
*simulation with the proposed procedure is also shown (see Section 3.2).*

|  | Flooded area (km$^2$) | Casualties[1] | Affected population[1] | Evacuated population[1] | Economic impact (M€) |
|---|---|---|---|---|---|
| Bosnia-Herzegovina | 266.3[1]; 831[2] | 25 | 1.6 million | 90000 | 2040 |
| Croatia | 53.5[1]; 110[3]; 210[4] | 3 | 38000 | 15000 | 300 |
| Serbia | 22.4[1] ; 221[3]; 350[5] | 51 | 1 million | 32000 | 1530[1] |

*Table 1. Summary of flood impacts at national level. Figures have been retrieved from the*
*following sources: 1 - ICPDR and ISRBC (2015); 2 - Bosnia-Herzegovina Mina Action Center*
*(BHMAC, Bajic et al 2015); 3 - Copernicus EMS Rapid Mapping Service; 4 - Wikipedia (2016);*
*5 - GeoSerbia geoportal (2016).*
### 3.2 Evaluation of the flood hazard mapping procedure
In our analysis we considered the river network of the Sava River basin, where some of the most
affected areas are located and for which detailed information is available from various reports.
To evaluate the skill of the flood hazard mapping procedure, we used observed flood magnitudes
(Figure 3) to identify the return period of peak discharges and thus select the appropriate flood
maps. In addition, we used the information on flood protection level and dyke failures to select
only those river sections where flooding actually occurred, either due to defence failures or
exceeding discharge. The resulting flood hazard map is referred to, for the remainder of this paper,
as the "reference simulation". Such a procedure excludes the uncertainty due to the hydrological
input from the analysis, focusing on the evaluation of the flood hazard mapping approach alone.
In other words, the test can be seen as an application of the procedure in the case of a single,
deterministic and "perfect" forecast. The resulting inundation map is displayed in Figure 4.
It is important to note that a margin of uncertainty remains because of the emergency measures
which were taken during the event. In several river sections of the Sava River, the flood defences
were actually able to withstand discharges well above their design value, thanks to timely
emergency measures such as heightening and strengthening of dykes. Moreover, the preparation
of temporary flood defences in the floodplains helped to protect some areas which would have
been otherwise flooded. A further feature of the methodology is that, where flood protections are
exceeded, flooding can occur on both river banks, while in the case of dyke failure flooding is
usually limited to one side where protection level is lower. This has not been corrected and
therefore the results are affected by this limitation.
The flood events in the Sava River have been mapped by several agencies and institutions using
both ground observations and satellite imagery (see UN SPIDER (2014) for a complete list). The
most comprehensive flood maps were developed by the Copernicus Emergency Management
System (EMS) using Sentinel-1 data (EMS, 2014), and by NASA using MODIS Aqua (UN
SPIDER, 2014). For Serbia, the Republic Geodetic authority has acquired and processed further
satellite images, which are available on the geoportal GeoSerbia (2016).
Despite these numerous available data sources, the evaluation of the simulated flood extent is not
straightforward. All of the available images were acquired when the flood was receding (from 19
May onwards), while flood peaks were observed between 15-17 May. Therefore, several areas
which have been reported as flooded in the available documentation are not included in the
detected flood footprints, which results in a significant difference between satellite-detected and
reported flood extent from ground surveys (see Table 1). On the other hand, EMS satellite maps
are designed to produce a low rate of false positive errors, and can therefore be considered as a
"lower limit" for the real flood extent. Finally, it must be considered that, for each country, the
available information sources report different extents of flooded area, as can be seen in Table 1.
In order to take these issues into account, we first compare the total simulated and reported flood
extent at country level, calculating over-estimation or under-estimation rates against all the
available reported data. Then, we evaluate the agreement between satellite-derived and simulated
flood extent considering those areas in the Sava River basin affected by the flood event and where
Copernicus satellite maps were available. Areas were grouped according to the main source of
flooding, either a tributary (e.g. Bosna River) or the Sava River. For the Sava River, we
considered two separate sectors because of the large extent of the flooded areas, and because flood
extent was not continuous. The agreement is evaluated using the hit ratio H (Alfieri et al., 2014b),
defined as:
$$H = (Fm \cap Fo)/(Fo) \times 100 \qquad\qquad (1)$$
where $Fm \cap Fo$ is the area correctly predicted as flooded by the model, and $Fo$ is the total
observed flooded area. Note that we did not consider indices to evaluate false hit ratios because,
as previously discussed, we know that the available satellite flood maps under-estimated the
actual flood extent. Consequently, false alarm ratio scores would be low without being supported
by reliable observations, giving an incorrect view of the performance. As a further element, we
compare the number of urban areas (cities, towns and villages) which were reported as flooded
by UNDAC (2014) and ICPDR and ISRBC (2015).

### 3.3 Evaluation of forecast-based flood hazard maps

To evaluate the overall performance of forecast-based flood hazard mapping, we considered the
EFAS forecasts issued on 12 and 13 May for the Sava river basin, i.e. immediately before the first
flood events occurred on 14 May. We first applied the standard procedure described in Section 2
above, to derive peak discharges, estimated return periods and flood maps using the median of
the EFAS ensemble forecasts. To provide a more complete overview of risk scenarios, we also
applied the procedure considering the 25 and 75 percentiles of discharge in the ensemble
forecasts. As a first step, we evaluate EFAS forecasts by comparing forecast and observed return
periods. Then, forecast-based flood hazard maps are evaluated against the reference simulation,
comparing the river sectors and the urban areas (or municipalities) at risk of flooding. Note that
we selected the reference simulation as the benchmark because it represents the best result
achievable in case of a perfect forecast. Conversely, we did not carry out a comparison against
observation-based flood maps, because they incorporate the effect of defence failures or
strengthening, which could only be considered as hypothetical scenarios in forecast-based maps.

### 3.4 Evaluation of impact assessment

Inundation maps derived from the reference simulation and flood forecasts have been used to
compute flood impacts in terms of number of affected people, affected major towns and cities,
and economic damage.
The results are compared with the available impact estimations both at national and local level.
For Serbia and Bosnia-Herzegovina, the national figures reported in Table 1 refer to the total
impact given by river floods, landslides and pluvial floods, and so cannot be directly compared
with methodology results. Therefore, the comparison has been done only for Croatia and for a
number of municipalities (e.g. Obrenovac in Serbia) where impacts can be attributed to river
flooding alone.
The figures for affected population computed with the reference simulation, are also useful to test
the reliability of the population map used as the exposure dataset. Similarly, damage estimations
provide an indication of the reliability of depth-damage curves for the study area.
As was done for the flood hazard maps, forecast-based risk estimations are evaluated against the
results from the reference simulation, comparing both population and damage figures. Note that
other variables produced by the operational procedure (e.g. roads affected, extent of flooded urban
and agricultural areas) could not be tested due to lack of observed data, and therefore are not
discussed here. To add a further term of comparison, affected population has been computed using
Copernicus EMS flood footprints.

## 382 *4) Results and discussions*


The results of the evaluation exercise are shown and discussed separately for each component of
the procedure.

### 386 *4.1 Flood hazard mapping*


Table 3 reports the observed flood extent data from available sources, and the simulated extent
derived from the reference simulation (i.e. the mapping procedure applied to discharge
observations). The ratios between simulations and observations are also included. Table 4 reports
the scores of the hit ratio (H) for the considered flooded sectors, together with a comparison of
towns flooded according to simulations and observations.

| | Flood extent (km$^2$) | | | |
|---|---|---|---|---|
| **Country** | Reference simulation | Satellite | Reported by ICPDR-ISRBC | Reported (other sources) |
| *Bosnia - Herzegovina* | 995 | 339 | 266.3 [1] | 831 [2] |
| *Croatia* | 919 (319) | 110 | 53.5 [1] | >210 [3] |
| *Serbia* | 582 | 221 | 22.4 [1] | >350 [4] |
| | Extent ratio | | | |
| **Country** | Reference simulation | Satellite | Reported by ICPDR-ISRBC | Reported (other sources) |
| *Bosnia - Herzegovina* | 1 | 0.34 | 0.27 | 0.84 |
| *Croatia* | 1 | 0.12 (0.34) | 0.06 (0.17) | >0.23 (0.66) |
| *Serbia* | 1 | 0.38 | 0.04 | >0.60 |


*Table 3. Comparison of observed and simulated flood extent data at country scale. Satellite*
*flood extent refers to Copernicus EMS maps. Values in parentheses for Croatia refer to a*
*modified simulation, as explained in the text. Reported flood extent has been retrieved from the*
*following sources: 1 - ICPDR and ISRBC (2015); 2 - Bosnia-Herzegovina Mina Action Center*
*(BHMAC, Bajic et al 2015); 3 - Wikipedia (2016); 4 - GeoSerbia geoportal (2016).*

| Affected areas | Hit ratio (H) | EMS flooded area (km$^2$) | Affected towns and cities |
|---|---|---|---|
| Bosna River | 90.6% | 58.46 | Maglaj, Doboj, Modriča |
| Sava River between confluences with Bosna and Drina | 63.9% | 134.76 | Orašje, Šamac, DonjiŽabar, Brcko, Gunja, (Zupanja), Bijeljina |
| Sava River between confluences with Drina  and Kolubara | 83.7% | 405.43 | Sabac, Obrenovac, Lazarevac |
| Total | 79.9% | 598.65 | |


Table 4. *Scores of the hit ratio (H) for the considered flooded sectors, and affected towns and*
*cities. Names in parentheses refer to towns and cities wrongly predicted as flooded, otherwise*
*towns and cities have been correctly predicted as flooded.*

As expected, the simulated flood extent is significantly larger, in all cases, than the satellite extent
(see Table 3), given the delay between the times of flood peak and image acquisition, as
mentioned in Section 3.2. Flood extent indicated in the ICPDR and ISRBC Report is also
consistently lower than values from both simulated and satellite maps.
On the other hand, simulated and reported extent are more comparable when considering data
reported by other sources. For Bosnia-Herzegovina, the simulated value is close to the reported
flood extent published in the report by Bajic et al. (2015). For Serbia, the flooded area detected
from GeoSerbia satellite maps is smaller than the simulation, but it has to be considered that these
maps have the same problem of delayed image acquisition as mentioned for the Copernicus maps.
For Croatia, the flood mapping methodology is largely over-estimating both the satellite-based
and reported flood extents. The main reason is that flooding on the left side of Sava was limited
due to the reinforcing of river dykes in the area close to the city of Zupanja, which could withstand
the reported 500-year return period discharge, despite having been designed for a one in one
hundred years event. In fact, all of the left bank of Sava in this area was reported as an area at risk
in case of a flood defence failure, and only the emergency measures taken prevented more severe
flooding (ICPDR and ISRBC, 2015). Therefore we performed an additional flood simulation
excluding any failure on the river's left bank between the Bosna confluence and Zupanja, and in
this case we found a total flood extent of 319 km$^2$. Even if this estimate still exceeds the reported
flood extent (Wikipedia, 2016), it has to be considered that this figure refers only to the Vukovar-
Srijem county, which was the most affected area, therefore the total affected area in the whole
country was probably larger.
Regarding Table 4, the scores of the hit ratio (H) indicate that the mapping procedure correctly
detected most of the flooded areas, with the partial exception of the lower Sava area. In particular,
the vast majority of towns reported to have been flooded are correctly detected by the simulations,
with only a few false alarms (e.g. the already mentioned Zupanja).
When looking at the results it is important to bear in mind the limitations of the procedure. As
mentioned in Section 2.3, the mapping is able to reproduce only maximum flood depths, while
the dynamics of the flood event are not taken into account. This means that processes like flood-
wave attenuation due to inundation occurring upstream, cannot be simulated, and possible flood
mitigation measures taken during the event are also not considered. Furthermore, due to the coarse
resolution (100 m) of the DEM used, flood simulations do not include small-scale topographic
features like minor river channels, dykes and road embankments.
## *4.2 Flood impact assessment*

Tables 5 summarizes reported and estimated impacts on population, based on both the reference
simulation and Copernicus satellite maps, for the three countries affected by floods in the Sava
basin. Tables 6 reports simulated and reported impacts on population for a number of
administrative regions where impacts can be attributed to floods only. For evaluating the
performance of impact assessment, we consider only Table 6, because national estimates in Table
5 include also people displaced by landslides and pluvial floods not simulated in EFAS.
Note that in both Tables we compare simulated impacts with figures for evacuated population
because reported estimates of affected population include also people affected by indirect effects
such as energy shortage and road blockage. Also, the figures for evacuated population are not
equivalent to directly affected population (i.e. whose houses were actually flooded). In some
areas, evacuation was taken as a precautionary measure, even if flooding did not eventually occur.
Conversely, not all the people living in flooded areas were evacuated after the event.

| Country | Evacuated population (reported) | Affected population (satellite) | Affected population (simulated) |
|---|---|---|---|
| Bosnia-Herzegovina | 90.000 | 51.010 | 215.200 |
| Croatia | 27.260 | 5.760 | 57.000 |
| Serbia | 32.000 | 13.700 | 29.800 |

*Table 5. Comparison of evacuated population (reported) and affected population estimated*
*from satellite and simulations in Bosnia-Herzegovina, Croatia and Serbia (source: ICPDR and*
*ISRBC, 2015).*

| Administrative area | Country | Evacuated population (reported) | Affected population (simulated) |
|---|---|---|---|
| Obrenovac municipality | Serbia | > 25,000 | 17,600 |
| Brcko district | Bosnia-H. | 1,200 | 1,700 |

| | | | |
|---|---|---|---|
| Brod-Posavina county | Croatia | 13,700 | 12,800 |
| Osjek-Baranja county | Croatia | 200 | 1,300 |
| Sisak-Moslavina county | Croatia | 2,400 | 3,300 |
| Požega-Slavonija county | Croatia | 2,300 | 1,500 |
| Vukovar-Srijem county | Croatia | 8,700 | 39,200 |


*Table 6. Comparison of evacuated population (reported) and affected population (simulated) in administrative areas in Bosnia-Herzegovina, Croatia and Serbia (source: ILO, 2014; ICPDR and ISRBC, 2015; Wikipedia, 2016)*


As can be seen, differences between results and reported figures are in the order of hundreds, suggesting that the procedure is able to provide a general indication of the impact on population, but with a limited precision where impacts are small, as in the case of Osjek-Baranja county. However, differences are larger for Vukovar-Srijem county in Croatia, and Obrenovac municipality in Serbia. For the former, this is due to over-estimation of flooded areas as discussed in Section 4.1. If dyke failures are not included in the simulation for this county, the affected population is reduced to 8,600 people, extremely close to the reported figure. The under-estimation in Obrenovac municipality may indicate that flood simulations are less reliable for urban areas, even if estimated figures still depict a major impact on the city. In fact, the DEM used in the simulations is mostly based on Shuttle Radar Topography Mission (SRTM) elevation data, known to be less accurate in urban and densely vegetated areas (Sampson et al., 2015).

For flood impacts related to monetary damage, the simulations for Croatia indicate a total damage of € 653 million, against a reported estimate of € 298 million. However, if the already mentioned over-estimation of flooded areas is considered, then the estimate decreases to € 190 million. The difference is relevant but still within the range of uncertainty of damage models quantified in previous studies (de Moel and Aerts, 2011; Wagenaar et al., 2016). As already mentioned, damage figures for Serbia and Bosnia-Herzegovina could not be used because available estimates aggregate damages from landslides and river and pluvial flooding.

The observed under-estimation should be evaluated considering the limitations of both observed data and damage assessment methodology. On the one hand, available damage functions for Croatia are not specifically designed for the country, as discussed in Section 2.3. Also, estimated damages include only direct damage to buildings, while infrastructural damage is only partially accounted for (e.g. damage to the dyke system). On the other hand, official estimates are affected by the absence of clear standards for loss assessment and reporting (Corbane et al., 2015; IRDR, 2015), and can strongly deviate from true extents and damages. Thieken et al. (2016) observed that reported losses are rarely complete and that it may be years before reliable loss estimates are available for an event.

*4.3EFAS forecasts*

Table 7 illustrates return periods of peak discharge derived from 12 and 13 May forecasts for the
main rivers of the Sava basin, visible in Figure 3. Simulations are compared against values
reported by ICPDR and ISRBC (2015).

| River | 12/5 25p. | 12/5 50p. | 12/5 75p. | 13/5 25p. | 13/5 50p. | 13/5 75p. | Reported |
|---|---|---|---|---|---|---|---|
| Return period forecast (years) | | | | | | | |
| Una | < 5 | < 5 | < 5 | < 5 | < 5 | < 5 | 50 |
| Sana | < 5 | < 5 | < 5 | < 5 | 5-10 | 5-10 | 50 |
| Bosna | < 5 | 5-10 | 10-20 | 5-10 | 20-50 | 50-100 | 500 |
| Vrbas | < 5 | 5-10 | 10-20 | 5-10 | 10-20 | 20-50 | 100 |
| Drina | < 5 | < 5 | 5-10 | <5 | 5-10 | 10-20 | 50 |
| Kolubara | 10-20 | 20-50 | 100-200 | 20-50 | 50-100 | >200 | 500 |
| Sava (upper reach) | < 5 | < 5 | < 5 | < 5 | < 5 | < 5 | 20 |
| Sava (middle reach) | < 5 | < 5 | < 5 | <5 | 5-10 | 5-10 | 500 |
| Sava (lower reach) | 5-10 | 5-10 | 10-20 | 10-20 | 10-20 | 20-50 | 100 |


*Table 7. Comparison of forecast and observed return periods in the main rivers of the Sava*
*Basin. The Sava River has been divided into three sectors. Upper: up to the confluence with the*
*Bosna River; Middle: between the confluences with Bosna and Drina rivers; Lower: from the*
*confluence with the Drina River to the confluence into the Danube River.*

Results show that the forecasts of 12 May are significantly far from observations even considering
the 75[th] percentile, with the exception of Kolubara River. The performance improves for the
forecasts of 13 May, when the magnitude of predicted discharges indicates a major flood hazard
in most of the considered rivers, although with a general under-estimation especially in the Una,
Sana and the upper and middle reaches of the Sava River. However, it has to be considered that
peak flow timing was rather variable across the Sava river basin, due to its extent. While in the
Kolubara river the highest discharges occurred on 14 and 15 May, peak flows in other tributaries
were reached later (between 14-16 May for Bosna River, on 16 May for Drina, on 17 May for
Sana River). On the main branch of the Sava River the flood peaks occurred after 17 May. Thus,
in a hypothetical scenario where EFAS risk forecasts were routinely used for emergency
management, on one hand there would have been still time to update flood forecasts, while on the
other hand, the forecast released on 13 May would have given emergency responders a warning
time of at least two days to plan response measures in several affected areas, chiefly in the
Kolubara and Bosna basins.
Figure 5 shows the inundation maps derived using the median of ensemble streamflow forecasts
issued on 12 and 13 May (i.e. the standard method adopted for the operational procedure).

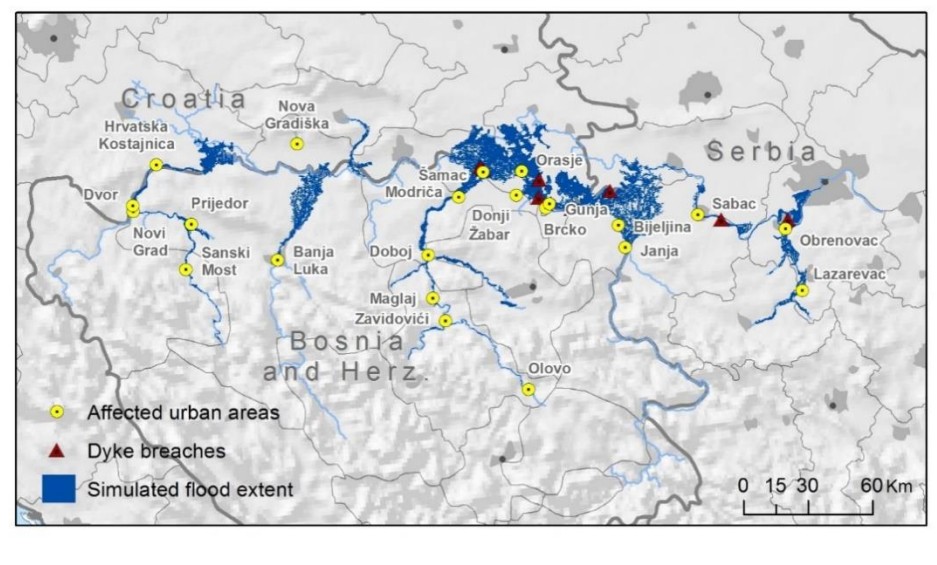

(a)


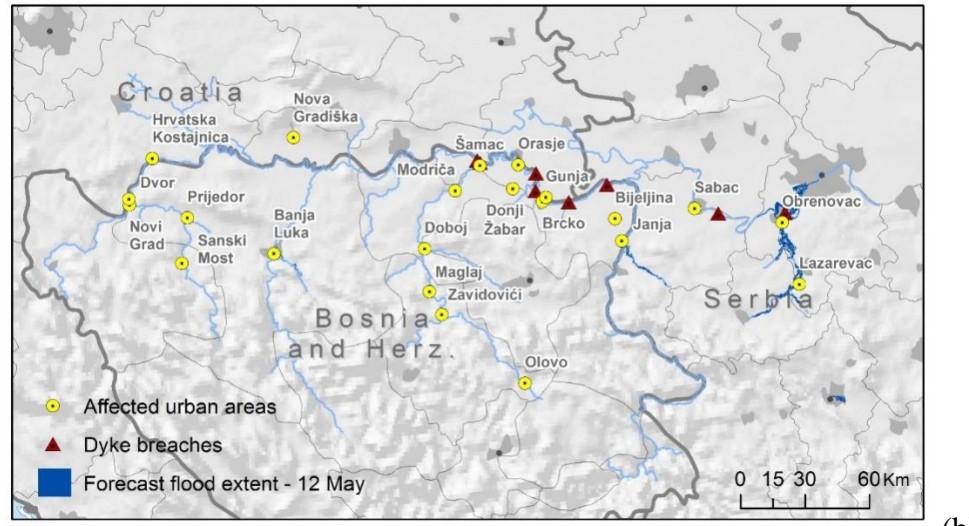

(b)


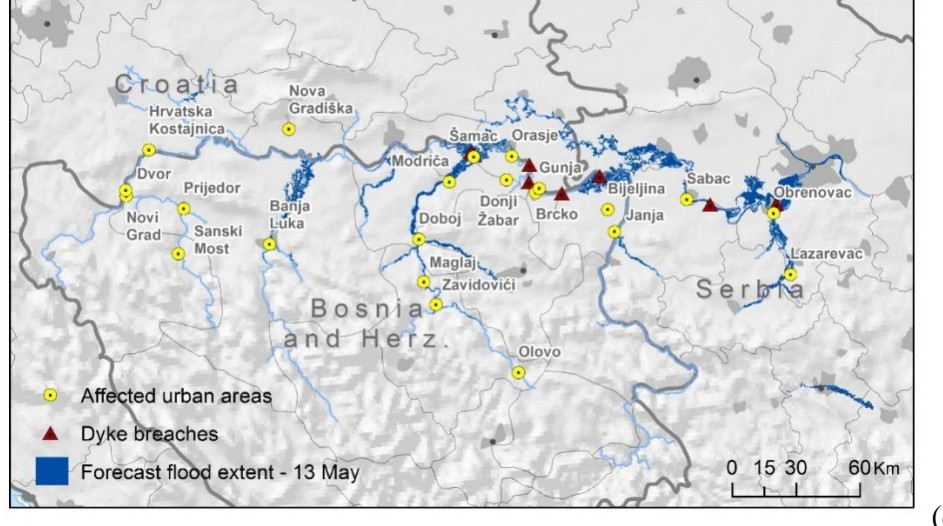

(c)

*Figure 5. (a) Simulated flood extent based on reference simulation; (b) 12 May forecast; (c) 13*
*May forecast. Locations of reported flooded urban areas and dyke failures are also shown.*
Furthermore, Table 8 illustrates the outcomes of impact forecasts, compared with impacts
obtained from the reference simulation. For both dates, we considered predicted maximum
streamflow values based on the $25^{th}$, $50^{th}$ and $75^{th}$ percentiles of the ensemble forecast. All
estimations are computed taking local flood protection levels into account.

| Country | 12/5 25p. | 12/5 50p. | 12/5 75p. | 13/5 25p. | 13/5 50p. | 13/5 75p. | Ref. Sim. |
|---|---|---|---|---|---|---|---|
| Flood extent (km$^2$) | | | | | | | |
| Bosnia-Herz. | 0 | 5 | 196 | 110 | 406 | 494 | 995 |
| Croatia | 0 | 0 | 100 | 54 | 95 | 135 | 919 |
| Serbia | 91 | 187 | 385 | 241 | 562 | 664 | 582 |
| affected population | | | | | | | |
| Bosnia-Herz. | 0 | 5,230 | 2,046 | 20,600 | 95,530 | 117,280 | 215,180 |
| Croatia | 0 | 0 | 3,600 | 1,940 | 2'780 | 4,480 | 57,050 |
| Serbia | 2,790 | 6,010 | 15,120 | 11,150 | 25,950 | 32,660 | 29,760 |
| Economic damage (million €) | | | | | | | |
| Bosnia-Herz. | 0 | 10 | 36 | 28 | 245 | 342 | 378 |
| Croatia | 0 | 0 | 41 | 13 | 22 | 37 | 653 |
| Serbia | 14 | 31 | 92 | 77 | 197 | 249 | 141 |

*Table 8. Comparison of forecast flood impacts with the reference simulation.*
The values in Table 8 allow the extension of the analysis done on predicted flood magnitudes,
and illustrates the evolution of flood risk depicted by EFAS ensemble forecasts. As can be seen,
the impact estimate derived from the 12 May forecast indicated a limited risk with the exception
of Serbia, even if the figures for the $75^{th}$ percentile already indicated the possibility of more
relevant impacts. The overall risk increases with the 13 May forecast, with severe and widespread
impacts associated to the ensemble forecast median, even though for Bosnia-Herzegovina and
especially Croatia there is still a significant under-estimation with respect to reference simulation.
A further important result is that the locations of forecast flooded areas are mostly consistent with
the reference simulation shown in Figure 3, with several urban areas already at risk of flooding
in the map based on the 13 May forecast (Figure 6).
In a hypothetical scenario, these results would have provided emergency responders with valuable
information to plan adequate counter-measures, based on the expected spatial and temporal
evolution of flood risk. A more detailed discussion on these topics is presented in Section 4.4.

## 4.4 Discussion

As mentioned in Section 1, the availability of a risk forecasting procedure able to transform hazard warning information into effective emergency management (i.e. risk reduction) (Molinari et al., 2013), opens the door to a wide number of new applications in emergency management and response. However, to better understand the limitations of such a procedure, as well as its potential for future applications, some considerations have to be made.

Firstly, it is important to remember that EFAS is a continental-scale system which is mainly designed to provide additional information and to support the activity of national flood emergency managers. Therefore, the practical use of risk forecasts to activate emergency measures would need to be discussed and coordinated with services and policy-makers at local level.

Secondly, the new procedure needs to undergo an accurate uncertainty analysis before risk forecasts can effectively be used for emergency management. While a detailed analysis is beyond the scope of this paper, to this end we have recently begun to evaluate the performance of the procedure for the flood events recorded in the EFAS and Copernicus EMS databases.

Another point to consider is the approach chosen to assess flood risk. In the current version of the procedure, we produce a single evaluation based on the ensemble forecast median, to provide a straightforward measure of the flood risk resulting from the overall forecast. A more rigorous approach would require analysis of all relevant flood scenarios resulting from EFAS forecasts, and estimation of their consequences together with the conditional probability of occurrence, given the range of ensemble forecast members and the forecast uncertainty (Apel et al., 2004). While such a framework would enable a cost-benefit analysis of response measures in an explicit manner, it would also require evaluation of the consequences of wrong forecasts, such as missing or under-estimating impending events, or issuing false alarms (Molinari et al., 2013; Coughlan et al., 2016). Given the difficulty of setting up a similar framework at a European scale, during the initial period of service the EFAS risk forecast will be used to plan "low regret" measures like satellite monitoring and warning of local emergency services. In the future, especially in areas where no local monitoring systems are available, EFAS risk forecasts may be used to plan more demanding measures such as monitoring of flood defences, deployment of emergency services and evacuation of endangered people. Even where local systems are operating, risk forecasts may provide additional, valuable information with respect to standard streamflow forecasts. However, in these areas emergency measures should be enacted on confirmation from local monitoring systems.

When designing the structure and output of risk assessment, it has to be considered that the type and amount of information provided must be based on requests from end-users. In fact, different end-users may be interested in different facets of flood impacts (Molinari et al., 2014), but at the same time it is important to avoid information overload during emergency management. Again, finding a compromise requires close collaboration with the user community.

For example, damage estimation has been included in the impact assessment at the request of EFAS end-users, despite the known limitations of the damage functions dataset, in particular the

absence of country-specific damage functions for the majority of countries in Europe. From this
point of view, the case study described in this paper is representative of the level of precision that
may be achieved in these countries. Future possible improvements include availability of detailed,
country-specific damage reports at building scale (i.e. reporting hazard variables and resulting
damage for different building categories), enabling the derivation of specific damage functions.
For similar reasons, this study has not addressed human safety and the protection of human life,
despite their importance in emergency management. The scale of application of the EFAS risk
assessment is not compatible with risk models for personal safety based on precise hydro-dynamic
analysis, such as that presented by Arrighi et al. (2016), whereas probabilistic risk methods (e.g.
de Bruijn et al., 2014) and the use of mortality rates calculated from previous flood events (e.g.
Jongman et al., 2015; Tanoue et al., 2016) are more feasible for integration, and these could be
tested for the next releases of the risk forecasting procedure.

## 5) Conclusions and next developments

This paper presents the first application of a risk forecasting procedure which is fully integrated
within a continental-scale flood early warning system. The procedure has been thoroughly tested
in all its components to reproduce the Sava River basin floods in May 2014, and the results
highlight the potential of the proposed approach.
The rapid flood hazard mapping procedure applied using observed river discharges, was able to
identify flood extent and flooded urban areas, while simulated impacts were comparable with
observed figures of affected population and economic damage. The evaluation was complicated
on the one hand by the scarcity of reported data at local scale, and on the other hand by the
considerable differences in impacts reported by different sources, especially regarding flood
extent. This is a well-recognised problem in flood risk literature, due to the fact that existing
standards for impact data collection and reporting are still rarely applied (Thieken et al., 2016).
Therefore, further improvements of impact models will require the availability of impact data
complying with international standards (Corbane et al., 2015; IRDR, 2015).
The use of EFAS ensemble forecasts enabled the identification of areas at risk with a lead-time
ranging from one to four days, and the correct evaluation of the magnitude of flood impacts,
although with some inevitable limitations, due to difference between simulated and observed
streamflow. When evaluating the outcomes, it is important to remember that, even in case of a
risk assessment based on "perfect" forecasts and modelling, simulated impacts will always be
different from actual impacts. As we have shown in the test case of the floods in the Sava River
basin, unexpected defence failures can occur for flow magnitudes lower than the design-level,
thus increasing flood impacts. On the other hand, flood defences might be able to withstand
greater discharges than their design-level, and emergency measures can improve the strength of
flood defences or create new temporary structures. Therefore, forecast-based risk assessment may
be regarded as plausible risk scenarios that can provide valuable information for local, national
and international authorities, complementing standard flood warnings. In particular, the explicit
quantification of impacts opens the way to more effective use of early warning information in
emergency management, enabling the evaluation of costs and benefits of response measures.
After a testing phase that started in September 2016, the procedure described in this paper has
been fully operational within the EFAS modelling chain, since March 2017. For the immediate
future, we plan to test a number of modifications and alternative approaches for the hazard
mapping and risk assessment components. For instance, flood hazard maps are now computed
using only the median of EFAS ensemble forecasts, but in principle the methodology can also be
applied to more ensemble members, in order to take account of (for example) flood scenarios that
are less probable but potentially more severe, and to provide a more complete risk evaluation
(such as the application described this paper). Furthermore, additional risk scenarios can be
produced, by considering the failure of local flood defences, or replacing EFAS flood hazard
maps with official hazard maps developed by national authorities, where available. The influence
of lead-time on flood predictions may also be assessed, for example by setting a criterion which
is based on forecast persistence over a period, to trigger the release of impact forecasts. All of
these alternatives will be tested in collaboration with the community of EFAS users, in order to
maximize the value of the information provided, and to avoid information overload, which can
be difficult to manage in emergency situations.
A further promising application which is being tested is the use of inundation forecasting to
activate rapid flood mapping from satellites, exploiting the European Commission's Copernicus
Emergency Mapping Service.
Finally, the proposed procedure will also be incorporated into the Global Flood Awareness
System (GloFAS), thereby enabling a near real-time flood risk alert system at a global scale.

## *Acknowledgements*


This study has been partially funded by the COPERNICUS programme and by an administrative
arrangement with Directorate General Humanitarian Aid and Civil Protection (DG ECHO) of the
European Commission.
The authors would like to thank Jutta Thielen, Vera Thiemig and Niall McCormick for their
valuable suggestions on the early versions of the manuscript.

## *Appendix*

## *Update of flood protection maps for Europe*

Table S1 shows a list of the updates to the flood protection level map developed by Jongman et
al. (2014), in use for the risk assessment procedure. The Table shows the rivers where values have
been updated, their geographic location (in some cases, protection values have been modified
only at specific locations along the river), previous and updated values, and the source of
information. Protection values are expressed in terms of years of an event's return period.
In addition to the modifications in Table S1, further updates of the EFAS database are planned,
using the global flood protection layer FloPROS (Scussolini et al., 2016).

| River | Region, Country | Previous values | Updated values | Reference |
|---|---|---|---|---|
| Sava | Croatia, Serbia, Bosnia-Herzegovina, | Not included -20 | 100 | ISRBC, 2014 |
| Drina | Bosnia-Herzegovina, | Not included | 50 | ISRBC, 2014 |
| Una, Vrbas, Sana, Bosna | Bosnia-Herzegovina, Croatia | Not included-10 | 30 | ISRBC, 2014 |
| Kolubara | Serbia | Not included | 50 | ISRBC, 2014 |

*Table S1. Update of the flood protection level map developed by Jongman et al. (2014), in use for*
*the risk assessment procedure.*

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
