# Peer review of "An operational procedure for rapid flood risk assessment in Europe"

_Natural Hazards and Earth System Sciences, 2016_

## Referee Comment (RC1) · Anonymous Referee #1 · 23 Nov 2016

The paper proposes a procedure for rapid flood impact assessment using a set of simulation models and a library of pre-processed flood inundation maps. Forecasted peak discharges are matched with corresponding flood maps from the library and mosaiced to provide a large-scale inundation map. This inundation map is then used to assess the impact of the flood in terms of population affected and economic damage.

The procedure is applied to the Balkan flood in May 2014 and the plausibility of the results is checked using observed and reported data. In this context also the limitations of the procedure are discussed. In view of an increasing importance of considering consequences within risk oriented flood management the paper addresses a relevant topic and could make a valuable contribution to the field. It is therefore suitable to be published in NHESS.

[Figure]

However, there are a number of points which should be taken into consideration to make the paper stronger. The most important ones are: 1) What is the definition of risk used in the paper? It would be more appropriate to use e.g. impact forecasting, particularly in the title and throughout the mansucript.

2) What is the benchmark you use? I think also this term is not very appropriate in the title because actually no benchmark is available. I would suggest to reword the title 'An operational procedure for rapid flood impact assessment in Europe'

3) The main achievement of the flood impact forecasts is currently not sufficiently elaborated. The focus shut be on the added value of the impact forecasts: i.e. the evaluation of consequences. Knowing the consequences of the flood in advance allows to take cost-benefit considerations into account which in turn allows to prioritize emergency and response measures. You should then also discuss issues concerning the protection of human life against economic loss.

4) Background information on different components of the system is sparse. For instance no information is given on the DEM used. Also, the model approach for flood impact assessment remains obscure. This should be clearly improved.

5) Figures 4, 5 and 6 should be combined in a multi panel graph for better comparison between the different settings.

Further remarks are given in the annotated PDF file.

Please also note the supplement to this comment:
http://www.nat-hazards-earth-syst-sci-discuss.net/nhess-2016-338/nhess-2016-338-RC1-supplement.pdf

―――――――――――――――――

**Supplement:**

[revised manuscript text omitted]

---

## Referee Comment (RC2) · Anonymous Referee #2 · 24 Jan 2017

The authors present a first attempt to develop a flood impact forecasting procedure that is fully integrated in a continental scale flood early warning system. They demonstrate this system by benchmarking various components against a flood events in May 2014 in Bosnia-Herzegovina, Croatia and Serbia. The paper builds on two directions of several previous works of the various authors: (1) the EFAS system that has previously been used for forecasting peak flows; and (2) the impact assessment module that has been used in several past risk studies for current and future conditions. In my opinion, this is a laudable effort – the need for such studies has been clearly vocalized in many past papers, and in many scientific and policy-related fora. I greatly appreciate the effort undertaken not simply to present the framework, but to try to benchmark it for an actual event. Of course, 1 event remains a limited benchmarking, but I believe that the benchmarking has been carried out in a way much more thorough to past studies

in large scale risk modelling. The novelty here is not in the models themselves, which have been developed in pervious papers, but bringing them together for impact forecasting. The paper is well written and clear, and provides enough level of detail on the already developed models, without too much repetition. I believe that the paper therefore is an important first step forward in this direction, and therefore merits publication in NHESS, subject to the authors being able to address the following issues:

Main comments 1. L119-121: "In case thresholds are exceeded persistently over several forecasts, flood warnings for the affected locations are issued to the members of the EFAS consortium." Please explain this statement better: which thresholds? And what is meant by "over several forecasts"? 2. L161-162: "We first identify the maximum discharge predicted over the full forecasting period, calculated using the median discharge from ensemble forecasts at each river grid cell". It is not clear to me from this sentence how this works. Do you take the maximum discharge across the entire ensemble for each lead time? (e.g. for lead time 1 day take the max discharge of all the ensemble members at 1 day lead) Or is something else meant here? Please clarify. 3. It is stated that the flood protection standards of Jongman et al. (2014) are used, and integrated with information from literature review and local authorities where available. In terms of transparency and reproducibility, I recommend a list (e.g Supplementary Information or in Appendix) showing the regions in which the values from Jongman et al were replaced, and which values were used. 4. In the validation of the inundation maps, the authors have chosen only to report the hit rates. I find this problematic, as a (theoretical) model that greatly overestimates flood extent would tend to have very high hit rates. Therefore, in itself it only tells half the story. I believe that it would be more prudent to also report the false alarm ratios. This is especially important, since in Table 3 it is shown that the simulations show a much larger flooded area than the observed datasets, which could be leading to the high hit rates. 5. With regards the validation of the flood risk (I think it would be better called "flood impacts"), expressed as affected population, on lines 414-415 it is stated that: ". . .results from the reference simulation match well figures reported for all the flooded counties of Croatia except for

the Vukovar-Srijem County." This is a very subjective statement: how is "match well" defined? For example, in the Osjek-Baranja Country, the observed dataset reports 200 people, whilst the simulated dataset suggests 1300 – i.e. a difference of 550%. I realise that the definitions used in the simulated/observed datasets are different, and so the direct comparison is difficult, but it would be more transparent to report the differences openly than disguise relatively large differences with ambiguous language. 6. One of the reasons given for the large difference in simulated damage between the reported and simulated dataset is that the damage curves applied have not yet been calibrated for Bosnia-Herzegovina, Croatia and Serbia. If this is the case, is it even useful to include this information in the warning? 7. In the conclusion, it is stated that the "Comparison of reported and simulated flooded areas suggests that the methodology enables to identify areas at risk well in advance..." Whilst the results do indeed show some encouraging skill, I think the phrase "well in advance" seems like oversell. The 12th May forecast for the 14th May flood showed little sign of flooding. The impacts were rather clear on the 13th May, giving a good confidence warning 1 day in advance. It is of course subjective whether 1 day is "well in advance" – it depends on the actions that planners need to take. Minor comments L60: the authors refer to a paper by Ward et al., 2016 to support the claim that "flood impact forecasts are increasingly being requested by end users of early warning systems". This facet is already discussed in Ward et al (2015), which would seem a more prudent paper to cite. L131: "we decided create" to "we decided to create" L179: Batista e Silva et al. (2012) → Batista and Silva et al. (2012)

L222: wide spread to widespread L368: "time o image" to "time of image"

References Ward, P.J., Jongman, B., Salamon, P., Simpson, A., Bates, P., De Groeve, T., Muis, S., Coughlan de Perez, E., Rudari, R., Trigg, M.A., Winsemius, H.C., 2015. Usefulness and limitations of global flood risk models. Nature Climate Change, 5, 712-715, doi:10.1038/nclimate2742.

---

## Author Comment (AC1) · 6 Mar 2017

**Reply to Reviewer 1**

The procedure is applied to the Balkan flood in May 2014 and the plausibility of the results is checked using observed and reported data. In this context also the limitations of the procedure are discussed. In view of an increasing importance of considering consequences within risk oriented flood management the paper addresses a relevant topic and could make a valuable contribution to the field. It is therefore suitable to be published in NHESS.

*We thank Reviewer 1 for his/her positive comments on our work.*

However, there are a number of points which should be taken into consideration to make the paper stronger. The most important ones are:

1) What is the definition of risk used in the paper? It would be more appropriate to use e.g. impact forecasting, particularly in the title and throughout the manuscript.

*In the manuscript we followed the standard definitions used in flood risk literature, that is, risk = hazard \* vulnerability \* exposure, and risk = probability \* consequences. However, it is true that in the first version of the manuscript the terms "impact" and "risk" were not correctly used. Therefore, we will carefully revise the use of these terms and recall their definition according to literature. In addition, we will discuss how the proposed procedure can be used for flood risk assessment (see the reply to point 'A" for more details).*

2) What is the benchmark you use? I think also this term is not very appropriate in the title because actually no benchmark is available. I would suggest to reword the title 'An operational procedure for rapid flood impact assessment in Europe'

*Following the Reviewer's suggestion, we will reword the title of the revised manuscript as 'An operational procedure for rapid flood risk assessment in Europe'.*

3) The main achievement of the flood impact forecasts is currently not sufficiently elaborated. The focus shut be on the added value of the impact forecasts: i.e. the evaluation of consequences. Knowing the consequences of the flood in advance allows to take cost-benefit considerations into account which in turn allows to prioritize emergency and response measures. You should then also discuss issues concerning the protection of human life against economic loss.

*We will address the remark raised by Reviewer 1 by expanding the analysis of possible uses of the new procedure, highlighting its added value in respect to standard flood forecasting. In particular, we will discuss how risk evaluation (i.e. considering both probability and consequence) can be used to develop cost-benefit analyses, including measures for human safety. In presenting this discussion we will keep a general perspective, because EFAS is a continental scale system and the practical design of any measure, including cost-benefit analyses, would need to be discussed and coordinated with emergency services and policy*

*makers at local level. Moreover, an accurate uncertainty analysis of EFAS risk forecasts is needed before developing practical applications (see also reply to point "A"), because the use of forecast for activating emergency measures require to take into account the possibility and consequences of acting "in vain", e.g. to issue a false alarm (Coughlan et al., 2016). To this end, we are currently to evaluate the procedure in past flood events recorded in the Copernicus database. As a first step, EFAS risk forecast could be used to activate "low regret" protection measures like monitoring of flood defence structures, warning of population and deployment of emergency services in areas at risk. All these considerations will be included in the revised version of the manuscript.*

*Regarding the issue of human safety, we will present a brief literature review of the methods for evaluating the risk of fatalities, and discuss the most feasible approaches for the proposed procedure, taking into account the issues previously mentioned. The scale of application of the EFAS risk assessment is not compatible with risk models for personal safety based on precise hydrodynamic analysis, like the one presented by Arrighi et al (2016), whereas probabilistic risk methods (e.g. de Bruijn et al., 2014) and the use of mortality rates calculated form previous flood events (e.g. Tanoue et al., 2016) are more feasible of integration. Again, these considerations will be included in the revised version of the manuscript.*

4) Background information on different components of the system is sparse. For instance no information is given on the DEM used. Also, the model approach for flood impact assessment remains obscure. This should be clearly improved.

*To address this remark, the revised manuscript will include more information on data and methods used in the study, including exhaustive references about the DEM and the flood impact assessment. For more details we refer to the replies to suggestions provided in the annotated PDF file (Points "G" to "O").*

5) Figures 4, 5 and 6 should be combined in a multi panel graph for better comparison between the different settings.

*In the revised version we will combine these figures in a single multi-panel graph, as suggested by the Reviewer.*

Further remarks are given in the annotated PDF file.

*Please find in the following the replies to all the remarks.*

P1: suggestion to change the order to be in accordance with previous clause.

*We will change the phrase as suggested by the Reviewer.*

a) P2 L47: a definition of how the term risk is used in this paper would be useful. The procedure proposed here provides a flood impact forecast. Flood risk (probability*consequences) is not assessed.

*As mentioned in the reply to Point 1, in the manuscript we followed the standard definitions used in flood risk literature, and in the revised manuscript we will carefully revise the use of the terms "impact" and "risk". To address the specific issue in L46-48, we will change the paragraph as follows: "While early warning systems are routinely used to predict flood magnitude, there is still a gap in the ability to translate flood forecasts into risk forecasts, that is, to evaluate the possible impacts generated by forecasted events, given their probability of occurrence (e.g. flood prone areas, affected population, flood damages losses). In addition, we will explain in detail why we used the definition "flood risk assessment" for the procedure presented in the manuscript. We reckon that the analysis of results has been mainly focused on impact assessment, whereas the evaluation of flood probability has not been addressed explicitly. However, the procedure does allow to compute the theoretical probability of occurrence of any forecasted event, because EFAS forecasts include the evaluation of discharge return periods at every point of the river network, based on the forecasted flood magnitude. To better illustrate this, in Section 4.3 of the revised manuscript we will evaluate EFAS forecast by comparing forecasted and observed return periods.*
*On this point, it should be considered that a correct risk evaluation for a forecast would require to estimate the conditional probability of occurrence given the flood forecast itself. For example, if the median of the EFAS ensemble forecast predicts a peak discharge of, say, 20 year return period, the probability that such a discharge will take place during the forecast period will be in theory higher that once every 20 years, depending on the situation and forecast reliability.*
*However, in order to be useful for emergency management, assessing conditional probability would require an accurate uncertainty analysis of the EFAS risk forecasting procedure, which is beyond the scope of this paper. We will report this discussion in a specific section of the revised paper, and mention the ongoing work aimed evaluating the procedure in past flood events recorded in the EFAS database.*

b) P2 L49: please provide context what is meant by static.

*We reckon that paragraph in L49-52 was not clearly written. It will be rewritten as follows: "Generally, flood impacts are evaluated considering reference risk scenarios where a fixed return period is used for all the area of interest, for instance based on official maps issued by competent authorities (EC 2007). However, this implies some degree of interpretation to delineate flood prone areas and define impacts in case of a flood forecast."*

c) P2 L57: check if repetition is needed

*We will delete the repetition as it is not necessary.*

d) P2 L60-65: One could argue that these tasks can already be done using flood forecasts. I think you should focus on the real added value of the impact forecast, which is the evaluation of consequences. Knowing the consequences of the flood in advance allows to take cost-benefit considerations into account which in turn allows to prioritize emergency and response measures. You should then also discuss protection of human life against economic loss.

*As discussed in the reply to Point 3, the revised manuscript will focus more on the added value given by evaluation of flood probabilities and consequences, highlighting the possibilities offered in respect to standard flood forecasting. Regarding the paragraph in L60-65, we will change it as follows: "At local scale, the joint evaluation of flood probabilities and consequences may not only increase preparedness of emergency services, but also allow cost-benefit considerations for planning and prioritizing response measures (e.g. strengthening flood defences, planning evacuation of people at risk). At European scale, the possibility to receive prior information on expected flood impacts would help the Emergency Response Coordination Centre (ERCC) in prioritizing and coordinating support to national emergency services."*

e) P2 L60: s.a. the term impact forecasting seems to be more appropriate than risk forecasting

*Please refer to our reply to Points 1 and A.*

f) P3 L100: only three components are introduced but four sub-sections are following. You should consider merging 2.1 and 2.2

*The separate description of the EFAS and map database was done to improve clarity. However, in order to keep consistency with the scheme in Figure 1, sections 2.1 and 2.2 will be changed into separate subsections 2.1.1 and 2.1.2.*

g) P4 L140: The reasoning behind this is not clear. Why don't you use the simulated hydrographs?

*The hydrographs simulated in the EFAS reference simulation are not referred to specific return periods, therefore we need to derive synthetic hydrographs for the return periods of interest. The extreme value analysis is used to derive peak discharge values for the return periods of interest, then we extract flow duration curves from the reference simulation which are used to design the shape of the synthetic hydrographs. Since the full procedure was described in Alfieri et al. (2014b) we did not provide a detailed description, however we will add these additional explanations in the revised manuscript.*

h) P4 L142: Background information about data sources, e.g. DEM should be added, since this is referred to later on L421

*The DEM used is a component of the River and Catchment Database developed at JRC and described in Vogt et al., (2007). We will include this reference in the revised manuscript.*

i) P5 L151: is it correct that only some river sections are shown?

*The conceptual representation is correct, however it must be noted that there is not a 1:1 correspondence between 5km and 100m river sections, given the different resolution. During the downscaling of discharge information, flood points are linked with the closest 0.1° pixel in the upstream direction where the coarse and high resolution river networks do not overlap. In particular, some 5km sections have no related sections in the 100m river network, while others can have more than one.*

j) P5 L163: Is this taken into account in the LISFLOOD-FP simulations in some way?

*We could not consider flood protections in LISFLOOD-FP simulations because we don't have information about the location and geometry of flood protection structures (e.g. levees). Therefore, LISFLOOD-FP simulations are run as if there were no protection structures.*

k) P6 L168: To which extend are these data available, for which fraction of river reaches from the whle network?

*Following a similar request from Reviewer 2, the revised paper will include an appendix with a list of the updates to the flood protection level map developed by Jongman et al. The list will show the regions where values have been updated, the old and new values, and the source of information.*

l) P6 L182: Please provide some background information on this approach.

*This information is taken from the map of World Cities available in the online ESRI database (http://www.arcgis.com/home/item.html?id=dfab3b294ab24961899b2a98e9e8cd3d).*

m) P6 L185: Please add a reference

*We will add a reference to the Corine Land Cover webpage on Copernicus website (http://land.copernicus.eu/pan-european/corine-land-cover).*

n) P6 L187: The references do not provide sufficient details about these depth-damage functions. The reference Huizinga 2007 is not a scientific publication and not available to the public. Additional information should be given here.

*In this study we used normalized damage functions which calculate the damage ratio as a function of water depth. Thus, damage fractions span from zero (no damage) to one (maximum damage). The damage ratio is then multiplied by the maximum damage value, calculated as a function of land use and country's GDP. Besides these additional details, in the revised paper we will refer to a recent JRC report by Huizinga et al. (2017), which describes a novel dataset of depth-damage functions at global scale, including the damage functions for Europe. The report will soon be publicly available.*

o) P6 L195: What is the approach to derive these additional curves? Please explain.

*In literature, country-specific depth-damage functions based on national data are available only for a limited number of countries. To produce a consistent dataset at European scale, Huizinga et al. (2007) elaborated available damage functions to derive averaged damage functions to be used for countries without specific functions. We therefore applied the same approach for Serbia and Bosnia-Herzegozina. More details can be found in the upcoming report by Huizinga et al. (2017).*

p) P7 L216: but in large areas of your test area additional damage curves have been derived, cf. L195, L223. What is this test worth for the European perspective?

*As stated in our previous reply (point "O"), the majority of European countries have no specific damage functions, therefore from this point of view the test is representative of the general data availability at European scale.*

q) P7 L236: Do you mean Sava river?

*The name is correct, the Sana River is a tributary of the Una River.*

r) P11 L310: please include references

*References to the ISRBC report (described in Section 3.1) will be included.*

s) P11 L321: But in the reference simulation also dike failures have been included in the inundation maps, right? cf L269

*True, but in real world applications it would not be possible to consider any information on what happened during the event, as forecast-based maps will be by definition produced in advance. Therefore we believe it is more correct to evaluate them without taking into account dyke failures or strengthening.*

t) P11 L335: you should introduce this scenario explicitly and explain on which information sources it is based.

*This scenario is actually the reference simulation described in Section 3.2, we will correct this.*

u) P12 L347: The term validation is not appropriate. You are rather doing plausibility checks on the different components of your system.

*We will use the term "evaluation" instead of validation in the revised paper.*

v) P12 L353: On which basis have these sections been selected? How many are considered out of the total number of sections?

*We used a confusing terminology here and we apologize for this. We considered here those areas in the Sava River basin affected by the flood event and where satellite flood extent maps from Copernicus were available. Areas were grouped considering the main source of flooding, either a tributary (e.g. Bosna) or the Sava River. For the Sava River, we considered two separate areas because of the large extent of the flooded areas, and because flood extent was not continuous. We could not consider other flooded areas for which satellite maps were not available.*

w) P12 Table 3: reference simulation

*This will be corrected as suggested.*

x) P12 L360: The footnotes could be aligned with Table 1.

*We will align footnotes as suggested.*

y) P12 L363: s.a. (*see above?*)

*This will be corrected as reported in the reply to Point "v".*

z) P13 L376: withstand

*Suggestion accepted.*

aa) P13 L392: no details provided on DEM, please add

*We will specify that the DEM has a 100m resolution.*

bb) P14 Table 6: simulated in reference simulation?

*Yes, this will be amended.*

cc) P14 L416: suggested to rephrase

*We will rephrase this in order to eliminate the repetition.*

dd) P14 L426: indicate or estimate

*We will replace "report" with "indicate".*

ee) P15 L430: but damage curves have been specifically derived for Serbia and Bosnia-Herzegovina (L195). This argument is therefore rather weak. How would such a calibration look like?

*The explanation on this point was not clear and we apologize for this. As reported in the reply to point "O", for Serbia and Bosnia- Herzegovina we applied depth-damage functions derived from data for other countries and averaged over all the European countries. However, the availability of detailed, country-specific damage reports at building scale (i.e. indicating the local water depth and the consequent damage for different building categories) would allow to derive specific damage functions.*

ff) P15 L433: You should also reflect on the completeness of official damage reports.

*We will elaborate on this point and make reference to the paper by Thieken et al. suggested by the Reviewer.*

gg) P15 L443: why? It would be interesting to see if the reference simulation is within the range of 25-75 quantiles.

*The revised paper will include results from the simulations of 25 and 75 quantiles for May 13.*

hh) P17 L476: please state how many days

*In the revised paper we will discuss with more details the performance regarding lead time. In fact, the timing of peak flow was variable across the Sava river basin, due to its extent. While in the Kolubara river the highest discharges occurred on $14^{th}$ and $15^{th}$ May, peak flows in other tributaries were reached later (between $14^{th}$ and $16^{th}$ for Bosna River, on $16^{th}$ for Drina, $17^{th}$ May for Sana River), and on the main branch of the Sava River the flood peaks occurred after $17^{th}$ May. Thus, the majority of affected areas the lead time was at least 2 days, if we consider the EFAS forecast issued on $13^{th}$ May. In the revised paper we will evaluate the performance considering these additional details, and discussing emergency actions that could be taken base on available lead time.*

ii) P17 L491: It would be valuable to refer to the existing international frameworks on impact data collection, see also: Thieken, A. H., Bessel, T., Kienzler, S., Kreibich, H., Müller, M., Pisi, S. and Schröter, K.: The flood of June 2013 in Germany: how much do we know about its impacts?, Nat. Hazards Earth Syst. Sci., 16(6), 1519–1540, doi:10.5194/nhess-16-1519-2016, 2016.

*We thank the Reviewer for the suggestion, in the revised paper we will elaborate on this point adding the suggested paper and further references from reports by IRDR (2015) and Corbane et al. (2015) on this topic.*

jj) P17 L496: please name the benefits

*Following the Reviewer's suggestion in Point 3, we will discuss how the proposed procedure allow to plan and prioritize response measures (e.g. strengthening and monitoring of flood defences, evacuation measures) based on cost-benefit considerations, leading to a considerable improvement in preparedness of emergency services.*

*Additional references*

*Arrighi. C., Oumeraci, H., Castelli, F., 2017. Hydrodynamics of pedestrians' instability in floodwaters. Hydrol. Earth Syst. Sci., 21, 515–531, 2017, doi:10.5194/hess-21-515-2017.*

*Corbane, C., de Groeve, T., and Ehrlich, D.: Guidance for Recording and Sharing Disaster Damage and Loss Data – Towards the development of operational indicators to translate the Sendai Framework into action, Report, JRC95505, EUR 27192 EN, 2015.*

*Coughlan de Perez, E. van Aalst, M. K. et al., Action-based flood forecasting for triggering humanitarian action, Hydrology and Earth System Sciences 20, 3549-3560, 2016. doi:10.5194/hess-20-3549-2016*

*De Bruijn, K. M., Diermanse, F. L. M., Beckers, J. V. L., An advanced method for flood risk analysis in river deltas, applied to societal flood fatality risk in the Netherlands .Nat. Hazards Earth Syst. Sci., 14, 2767–2781, 2014, doi:10.5194/nhess-14-2767-2014.*

*ESRI map of World Cities, accessed on 06/03/2017 at http://www.arcgis.com/home/item.html?id=dfab3b294ab24961899b2a98e9e8cd3d.*

*European Commission, Copernicus Land Monitoring Service, accessed on 02/02/2017 at http://land.copernicus.eu/pan-european/corine-land-cover.*

*IRDR – Integrated Research on Disaster Risk: Guidelines on Measuring Losses from Disasters: Human and Economic Impact Indicators, Integrated Research on Disaster Risk, Beijing, IRDR DATA Publication No. 2, 2015.*

*Huizinga, J., de Moel, H., Szewczyk, W., Global flood damage functions, JRC Technical Reports, in publication.*

*Tanoue, M., Hirabayashi, Y., Ikeuchi, H., Global-scale river flood vulnerability in the last 50 years. Scientific Reports 6 (2016).*

*Vogt et al., A pan-European river and catchment database, JRC Reference Reports 2007, doi:0.2788/35907*

---

## Author Comment (AC2) · 6 Mar 2017

**Reply to Reviewer 2**

The authors present a first attempt to develop a flood impact forecasting procedure that is fully integrated in a continental scale flood early warning system. They demonstrate this system by benchmarking various components against a flood events in May 2014 in Bosnia-Herzegovina, Croatia and Serbia. The paper builds on two directions of several previous works of the various authors: (1) the EFAS system that has previously been used for forecasting peak flows; and (2) the impact assessment module that has been used in several past risk studies for current and future conditions. In my opinion, this is a laudable effort – the need for such studies has been clearly vocalized in many past papers, and in many scientific and policy-related fora. I greatly appreciate the effort undertaken not simply to present the framework, but to try to benchmark it for an actual event. Of course, 1 event remains a limited benchmarking, but I believe that the benchmarking has been carried out in a way much more thorough to past studies in large scale risk modelling. The novelty here is not in the models themselves, which have been developed in pervious papers, but bringing them together for impact forecasting. The paper is well written and clear, and provides enough level of detail on the already developed models, without too much repetition.

*We thank Reviewer 1 for his/her positive comments on our work.*

I believe that the paper therefore is an important first step forward in this direction, and therefore merits publication in NHESS, subject to the authors being able to address the following issues:

1) L119-121: "In case thresholds are exceeded persistently over several forecasts, flood warnings for the affected locations are issued to the members of the EFAS consortium." Please explain this statement better: which thresholds? And what is meant by "over several forecasts"?

*The thresholds mentioned are the local discharge values corresponding to 1, 2, 5 and 20-year return periods, calculated from the EFAS reference simulation (L 113-117). The persistence criterion is that the 5 year threshold must be exceeded over 3 consecutive forecasts.  To clarify this part, lines 117-121 will be rephrased as follows:*
*"The reference simulation is also used to estimate discharge values for the return periods corresponding to 1, 2, 5 and 20-year at every point of the river network. All flood forecasts are compared against these thresholds and the threshold exceedance calculated. In case the 5 year threshold is consistently exceeded over 3 consecutive forecasts, flood warnings for the affected locations are issued to the members of the EFAS consortium. The persistence criterion has been introduced to reduce the number of false alarms and focus on large fluvial floods caused mainly by widespread severe precipitation, combined rainfall with snow-melting or prolonged rainfalls of medium intensity".*

2) L161-162: "We first identify the maximum discharge predicted over the full forecasting period, calculated using the median discharge from ensemble forecasts at each river grid cell". It is not clear to me from this sentence how this works. Do you take the maximum discharge

across the entire ensemble for each lead time? (e.g. for lead time 1 day take the max discharge of all the ensemble members at 1 day lead) Or is something else meant here? Please clarify.

*We first consider the median of the ensemble forecast, and then select the maximum discharge of the median over the full forecasting period (10 days). In the revised paper we will rewrite this sentence accordingly.*

3) It is stated that the flood protection standards of Jongman et al. (2014) are used, and integrated with information from literature review and local authorities where available. In terms of transparency and reproducibility, I recommend a list (e.g Supplementary Information or in Appendix) showing the regions in which the values from Jongman et al were replaced, and which values were used.

*Following the Reviewer's suggestion, the revised paper will include an appendix with a list of the updates and additions to the flood protection level map developed by Jongman et al. The list will show the regions where values have been updated, the old and new values, and the source of information.*

4) In the validation of the inundation maps, the authors have chosen only to report the hit rates. I find this problematic, as a (theoretical) model that greatly overestimates flood extent would tend to have very high hit rates. Therefore, in itself it only tells half the story. I believe that it would be more prudent to also report the false alarm ratios. This is especially important, since in Table 3 it is shown that the simulations show a much larger flooded area than the observed datasets, which could be leading to the high hit rates.

*We agree with the Reviewer on that presenting the results also in term of overestimation is necessary. To this end, in the revised version Table 3 will include overestimation (or underestimation) ratios between simulations and all the available observations, to provide a more objective presentation of the results.*
*However, regarding the results in Table 4 we believe that it is more correct not to compute false hit ratio because, as discussed in the manuscript, we know that the available satellite flood maps underestimated the actual flood extent. As such, false alarm ratio scores would be low without being supported by reliable observations, giving an incorrect view of the performance.*

5) With regards the validation of the flood risk (I think it would be better called "flood impacts"), expressed as affected population, on lines 414-415 it is stated that: ". . .results from the reference simulation match well figures reported for all the flooded counties of Croatia except for the Vukovar-Srijem County." This is a very subjective statement: how is "match well" defined? For example, in the Osjek-Baranja Country, the observed dataset reports 200 people, whilst the simulated dataset suggests 1300 – i.e. a difference of 550%. I realise that the definitions used in the simulated/observed datasets are different, and so the direct comparison is difficult, but it would be more transparent to report the differences openly than disguise relatively large differences with ambiguous language.

*We agree on that the evaluation of results requires the use of a more precise language. In the revised manuscript, we will present both absolute and relative differences between observations and simulations, in order to provide a more objective discussion of results, and we will avoid ambiguous terms.*
*Also, we will carefully revise the use of terms "flood risk" and "flood impact" in the paper (see also the reply to Reviewer 1 for a more detailed discussion on this point).*

6) One of the reasons given for the large difference in simulated damage between the reported and simulated dataset is that the damage curves applied have not yet been calibrated for Bosnia-Herzegovina, Croatia and Serbia. If this is the case, is it even useful to include this information in the warning?

*The operational rapid risk assessment includes damage estimation because of specific requests of EFAS end users, therefore we deemed correct to show the results for the case study here presented, even if available data for validation are limited to Croatia and no country-specific damage functions are available. Moreover, from this point of view the test area is representative of the majority of European countries, which have not specific damage functions. Even considering the mentioned issues, we think that the application can provide useful information on the performance of the modelling framework.*

7) In the conclusion, it is stated that the "Comparison of reported and simulated flooded areas suggests that the methodology enables to identify areas at risk well in advance. . ." Whilst the results do indeed show some encouraging skill, I think the phrase "well in advance" seems like oversell. The 12th May forecast for the 14th May flood showed little sign of flooding. The impacts were rather clear on the 13th May, giving a good confidence warning 1 day in advance. It is of course subjective whether 1 day is "well in advance" – it depends on the actions that planners need to take.

*We apologize for not having been precise on presenting the performance regarding lead time. In fact, the timing of peak flow was rather variable across the Sava river basin, due to its extent. While in the Kolubara river the highest discharges occurred on 14$^{th}$ and 15$^{th}$ May, peak flows in other tributaries were reached later (between 14$^{th}$ and 16$^{th}$ for Bosna River, on 16$^{th}$ for Drina, 17$^{th}$ May for Sana River), and on the main branch of the Sava River the flood peaks occurred after 17$^{th}$ May. Thus, for the majority of affected areas the lead time was at least 2 days, if we consider the EFAS forecast issued on 13$^{th}$ May. In the revised paper we will evaluate the performance considering these additional details, and discussing emergency actions that could be taken base on available lead time.*

Minor comments:

a) L60: the authors refer to a paper by Ward et al., 2016 to support the claim that "flood impact forecasts are increasingly being requested by end users of early warning systems". This facet is already discussed in Ward et al (2015), which would seem a more prudent paper to cite.

*We agree with the Reviewer, in the revised manuscript the reference will be replaced as suggested.*

b) L131: "we decided create" to "we decided to create" ; L222: wide spread to widespread; L368: "time o image" to "time of image".

*These typos will be corrected.*

c) L179: Batista e Silva et al. (2012) → Batista and Silva et al. (2012)

*The reference is actually correct, first author's surname is "Batista e Silva".*

---

## Author Response (AR1)

**Author's response**

Please find below the point-by-point response to the reviews, which includes all relevant changes made in the manuscript in respect to the first version. Note that with respect to Author's responses published during the discussion phase, we provided additional explanations and we slightly modified some replies, in accordance to the final modifications made to the manuscript.
The response to the reviews is followed by the marked-up manuscript version.

**Reply to Reviewer 1**

The procedure is applied to the Balkan flood in May 2014 and the plausibility of the results is checked using observed and reported data. In this context also the limitations of the procedure are discussed. In view of an increasing importance of considering consequences within risk oriented flood management the paper addresses a relevant topic and could make a valuable contribution to the field. It is therefore suitable to be published in NHESS.

*We thank Reviewer 1 for his/her positive comments on our work.*

However, there are a number of points which should be taken into consideration to make the paper stronger. The most important ones are:

1) What is the definition of risk used in the paper? It would be more appropriate to use e.g. impact forecasting, particularly in the title and throughout the mansucript.

*We reckon that in the first version of the manuscript the terms "impact" and "risk" were not correctly used. We carefully revised the use of these terms through the text following the standard definitions used in flood risk literature (see for instance page 2, lines 48-50).*
*In particular, we modified Section 2 to clarify how the proposed procedure provides all the elements for evaluating flood risk, following the definition risk = hazard \* vulnerability \* exposure (recalled at page 3, lines 96-97). Section 2.2.2 explains how EFAS ensemble discharge forecasts are elaborated to estimate the expected flood hazard, thus taking into account the probability of occurrence of the forecast flood event. See also the reply to point 'A' for additional details.*

2) What is the benchmark you use? I think also this term is not very appropriate in the title because actually no benchmark is available. I would suggest to reword the title 'An operational procedure for rapid flood impact assessment in Europe'

*Following the Reviewer's suggestion, we reworded the title as 'An operational procedure for rapid flood risk assessment in Europe'*

3) The main achievement of the flood impact forecasts is currently not sufficiently elaborated. The focus shut be on the added value of the impact forecasts: i.e. the evaluation of consequences. Knowing the consequences of the flood in advance allows to take cost-benefit considerations into account which in turn allows to prioritize emergency and response measures. You should then also discuss issues concerning the protection of human life against economic loss.

*We addressed this remark raised by Reviewer 1 by adding a new discussion section in the revised paper (4.4), where we analyse the potential uses of the new procedure and discuss current limitations that need to be overcome before the full potential of risk forecasts can effectively be used for emergency management. This is addressed in particular in the following part (pages 20-21, lines 581-629): "As discussed in the Introduction, the availability of a risk forecasting procedure able to transform hazard warning information into effective emergency management (i.e. risk reduction) (Molinari et al., 2013), opens the door to a wide number of new applications in emergency management and response. However, to better understand the limitations of the procedure, as well as its potential for future applications, some considerations have to be made.*
*First, it is important to remember that EFAS is a continental scale system which is mainly designed to provide additional information and support the activity of national flood emergency managers. Therefore, the practical use of risk forecasts to activate emergency measures would need to be discussed and coordinated with services and policy makers at local level.*
*Second, the new procedure needs to undergo an accurate uncertainty analysis before risk forecasts can effectively be used for emergency management. While a detailed analysis is beyond the scope of this paper, to this end, we recently started to evaluate the performance of the procedure by applying it to flood events recorded in the EFAS and Copernicus EMS databases.*
*Another point to consider is the approach chosen to assess flood risk. In the current version of the procedure, we decided to produce a single evaluation based on the ensemble forecast median to provide a straightforward measure of the flood risk resulting from the overall forecast. A more rigorous approach would require to analyse all relevant flood scenarios resulting from EFAS forecasts and estimate their consequences together with the conditional probability of occurrence, given the flood forecast itself (e.g. the range of ensemble forecast) and forecast uncertainty (Apel et al., 2004).While such a framework would enable the analysis*

*of benefits and costs of response measures in an explicit manner, it would also require to evaluate the consequences of wrong forecasts, like missing or underestimating impending events ,or issuing false alarms (Molinari et al., 2013; Coughlan et al., 2016). Given the difficulty of setting up a similar framework at European scale, during the initial period of service EFAS risk forecast will be used to plan "low regret" measures like satellite monitoring and warning of local emergency services. For instance, we are currently evaluating the use of EFAS risk forecast to trigger satellite rapid flood mapping through Copernicus EMS, with the aim of improving response time and detection of flooded areas. More demanding measures (e.g. monitoring flood defences, deployment of emergency services in areas at risk planning evacuation of people at risk), could instead be put in place upon confirmation from local flood monitoring systems".*

*The issue of human safety is not addressed in the current version of the EFAS risk assessment procedure because this information has not been requested so far by end users. However, in section 4.4 we also discussed this issue and its possible inclusion in page 21, lines 587-593): "For the same reasons, human safety and the protection of human life have not been addressed in this study, despite their importance in emergency management. The scale of application of the EFAS risk assessment is not compatible with risk models for personal safety based on precise hydrodynamic analysis, like the one presented by Arrighi et al. (2016), whereas probabilistic risk methods (e.g. de Bruijn et al., 2014) and the use of mortality rates calculated from previous flood events (e.g. Tanoue et al., 2016) are more feasible of integration and could be tested for next releases of the risk forecasting procedure".*

4) Background information on different components of the system is sparse. For instance no information is given on the DEM used. Also, the model approach for flood impact assessment remains obscure. This should be clearly improved.

*Following the Reviewer's remark, the revised manuscript now includes more information on data and methods used in this study, including exhaustive references about the DEM and flood impact assessment. For more details we refer to the replies to points "G" to "O".*

5) Figures 4, 5 and 6 should be combined in a multi panel graph for better comparison between the different settings.

*In the revised version we combined these figures in a single multi-panel graph, as suggested by the Reviewer.*

Further remarks are given in the annotated PDF file.

*Please find in the following a reply to all the remarks.*

P1: suggestion to change the order to be in accordance with previous clause.

*We changed the phrase as suggested by the Reviewer.*

a) P2 L47: a definition of how the term risk is used in this paper would be useful. The procedure proposed here provides a flood impact forecast. Flood risk (probability*consequences) is not assessed.

*As mentioned in the reply to Point 1, in the revised manuscript we carefully revised the use of the terms "impact" and "risk". To begin with, we explicitly introduced a definition of flood risk as suggested by Reviewer 1 in page 2, lines 46-49: "While early warning systems are routinely used to predict flood magnitude, there is still a gap in the ability to translate flood forecasts into risk forecasts, that is, to evaluate the possible consequences generated by forecast events (e.g. flood prone areas, affected population, flood damages losses), given their probability of occurrence."*
*In addition, in the revised manuscript we modified Section 2 to clarify how the proposed procedure provides all the elements for evaluating flood risk, following the definition risk = hazard * vulnerability * exposure. Section 2.2.2 explains how EFAS ensemble discharge forecasts are elaborated to estimate the expected flood hazard, thus taking into account the probability of occurrence of the forecast flood event. Furthermore, in Section 4.3 of the revised manuscript we now provide an additional analysis of EFAS forecasts by comparing forecasted and observed return periods, to evaluate whether predicted flood hazard resulting from ensemble members is comparable with observations. Finally, in Section 4.4 we provide additional discussion on the approach chosen to quantify flood risk based on EFAS ensemble forecasts (see reply to point 1 for more details).*

b) P2 L49: please provide context what is meant by static.

*We rewrote that paragraph, which now reads as follows (page 2, lines 49-52): "Generally, flood impacts are evaluated considering reference risk scenarios where a fixed return period is used for all the area of interest, for instance based on official maps issued by competent authorities (EC 2007). However, this implies some degree of interpretation to delineate flood prone areas and define impacts in case of a flood forecast."*

c) P2 L57: check if repetition is needed

*We deleted the repetition as it was not necessary.*

d) P2 L60-65: One could argue that these tasks can already be done using flood forecasts. I think you should focus on the real added value of the impact forecast, which is the evaluation of consequences. Knowing the consequences of the flood in advance allows to take cost-benefit considerations into account which in turn allows to prioritize emergency and response measures. You should then also discuss protection of human life against economic loss.

*As discussed in the reply to Point 3, the revised manuscript will focus more on the added value given by evaluation of flood probabilities and consequences, highlighting the possibilities offered in respect to standard flood forecasting. Regarding the paragraph considered by Reviewer 1, we modified it as follows (page 2, lines 60-65) :"At local scale, the joint evaluation of flood probabilities and consequences  may not only increase preparedness of emergency services, but also allow cost-benefit considerations for planning and prioritizing response measures (e.g. strengthening flood defences, planning evacuation of people at risk). At European scale, the possibility to receive prior information on expected flood risk would help the Emergency Response Coordination Centre (ERCC) in prioritizing and coordinating support to national emergency services."*

e) P2 L60: s.a. the term impact forecasting seems to be more appropriate than risk forecasting

*Please refer to our reply to Point 1 and A.*

f) P3 L100: only three components are introduced but four sub-sections are following. You should consider merging 2.1 and 2.2

*In order to keep consistency with the scheme in Figure 1, sections 2.2 and 2.3 in the first version of the manuscript have been placed into separate subsections 2.2.1 and 2.2.2. Note that we kept separated the descriptions of the map database and rapid flood mapping for the sake of clarity.*

g) P4 L140: The reasoning behind this is not clear. Why don't you use the simulated hydrographs?

*To clarify this part, we added the following paragraph in page 5, lines 145-150: "Since hydrographs simulated in the EFAS reference simulation are not referred to specific return periods, we use a statistical analysis of extreme values to derive peak discharges in every cell of the river network for reference return periods of 10, 20, 50, 100, 200 and 500 years. In addition, we extract flow duration curves from the reference simulation which are used together with peak discharges to calculate synthetic flood hydrographs (see Alfieri et al., 2014b for a detailed description"*

h) P4 L142: Background information about data sources, e.g. DEM should be added, since this is referred to later on L421

*The DEM used for downscaling the river network and running flood simulations is a component of the River and Catchment Database developed at JRC and described in Vogt et al., (2007). This reference was included in the revised manuscript (page 5, lines 158-163).*

i) P5 L151: is it correct that only some river sections are shown?

*The conceptual representation is correct, however it must be noted that there is not a 1:1 correspondence between 5km and 100m river sections, given the different resolutions. During downscaling of discharge information, where the coarse and high resolution river networks do not overlap, flood points are linked with the closest 0.1° pixel in the upstream direction. In particular, some 5km sections have no related section in the 100m river network, while others can have more than one. This additional explanation has been added in page 5, lines 151-158.*

j) P5 L163: Is this taken into account in the LISFLOOD-FP simulations in some way?

*We could not consider flood protections in LISFLOOD-FP simulations because we don't have information about the location and geometry of flood protection structures (e.g. levees). Therefore, LISFLOOD-FP simulations are run as if there were no protection structures. This additional explanation has been added in page 7, lines 190-193.*

k) P6 L168: To which extend are these data available, for which fraction of river reaches from the whole network?

*Following a similar request from Reviewer 2, the revised paper now includes an appendix with a list of the updates to the flood protection level map developed by Jongman et al.
The list shows the regions where values have been updated, the old and new values, and the source of information.*

l) P6 L182: Please provide some background information on this approach.

*This information is taken from the map of World Cities available in the online ESRI database. The reference is now reported in the revised manuscript.*

m) P6 L185: Please add a reference

*We added in the revised manuscript a reference to the Corine Land Cover webpage on Copernicus website (http://land.copernicus.eu/pan-european/corine-land-cover)*

n) P6 L187: The references do not provide sufficient details about these depth-damage functions. The reference Huizinga 2007 is not a scientific publication and not available to the public. Additional information should be given here.

*In the revised manuscript (page 7, lines 213-221) we have provided additional information on the depth-damage functions used: "More specifically, we use a set of normalized damage functions to calculate the damage ratio as a function of water depth, spanning from zero (no damage) to one (maximum damage). The damage ratio is then multiplied by the maximum damage value, calculated as a function of land use and country's GDP, to calculate actual damage. Separate damage functions are applied for the land use classes that are more vulnerable to flooding (residential, commercial, industrial, agricultural). In addition, to account for the variable value of assets within one country, damage values are corrected considering the ratio between the gross domestic product (GDP) of regions (identified according to the Nomenclature of Territorial Units for Statistics (NUTS), administrative level 1) and country's GDP."*
*Besides these additional details, in the revised paper we added a reference to a recent JRC report by Huizinga et al. (2017), which describes a novel dataset of depth-damage functions at global scale, including also the damage functions for Europe. This report will soon be publicly available.*

o) P6 L195: What is the approach to derive these additional curves? Please explain.

*We have added additional details on this point in page 7, lines 222-228: "For countries where specific damage functions could be found in literature, Huizinga et al. (2007) produced normalized functions based on this national data. In addition, the same authors elaborated averaged functions to be used for countries without national data, in order to produce a consistent dataset at European scale. The same approach has been applied in the present study to elaborate damage curves for countries not included in the original database, like Serbia and Bosnia-Herzegovina .The complete set of damage functions and the detailed description of the methodology are available as supplementary data of the recent report by Huizinga et al. (2017)"*

p) P7 L216: but in large areas of your test area additional damage curves have been derived, cf. L195, L223. What is this test worth for the European perspective?

*This comment has been addressed in page 21, lines 580-587:" … damage estimation has been included in the impact assessment upon request of EFAS end users, despite the known*

*limitations of the damage functions dataset, in particular the absence of country-specific damage functions for the majority of countries in Europe. From this point of view, the case study described in this work is representative of the level of precision that might be achievable in these countries. Future improvements can be possible with the availability of detailed, country-specific damage reports at building scale (i.e. reporting hazard variables and the consequent damage for different building categories) would allow to derive specific damage functions.*

q) P7 L236: Do you mean Sava river?

*No it is correct, the Sana River is a tributary of the Una River.*

r) P11 L310: please include references

*We included a reference to the ISRBC report (page 12, line 353).*

s) P11 L321: But in the reference simulation also dike failures have been included in the inundation maps, right? cf L269

*True, but in forecast-based maps the effect of defence failures or strengthening could be considered only as hypothetical scenarios. Therefore we deemed more correct to evaluate them without taking into account dyke failures or strengthening. These considerations have been included in page 12, lines 355-359.*

t) P11 L335: you should introduce this scenario explicitly and explain on which information sources it is based.

*This scenario is actually the reference simulation described in Section 3.2, we corrected this oversight.*

u) P12 L347: The term validation is not appropriate. You are rather doing plausibility checks on the different components of your system.

*In the revised paper we used the term "evaluation" instead of validation (page 13 line 382).*

v) P12 L353: On which basis have these sections been selected? How many are considered out of the total number of sections?

*We used a confusing terminology here and we apologize for this. We considered here those areas affected during the flood event in the Sava River where satellite flood extent maps from*

*Copernicus were available. Areas were grouped considering the main source of flooding, either a tributary (e.g. Bosna) or the Sava River. For the Sava River, we considered two separate areas because of the large extent of the flooded areas, and because flood extent was not continuous. We could not consider other flooded areas for which satellite maps were not available. This explanation has been included in pages 11-12, lines 328-333.*

w) P12 Table 3: reference simulation

*Table 3was corrected as suggested*

x) P12 L360: The footnotes could be aligned with Table 1.

*We aligned footnotes as suggested.*

y) P12 L363: s.a. (*see above?*)

*We corrected this as reported in the reply to Point "v"*

z) P13 L376: withstand

*Suggestion accepted*

aa) P13 L392: no details provided on DEM, please add

*In the revised manuscript it is specified now that the DEM has a 100m resolution (see also Point 4 and h).*

bb) P14 Table 6: simulated in reference simulation?

*Yes, this has been amended.*

cc) P14 L416: suggested to rephrase

*We rephrased the sentence to eliminate the repetition.*

dd) P14 L426: indicate or estimate

*We replaced "report" with "indicate".*

ee) P15 L430:but damage curves have been specifically derived for Serbia and Bosnia-Herzegovina (L195). This argument is therefore rather weak. How would such a calibration look like?

*The explanation on this point was not clear and we apologize for this. As explained in the reply to point "O", for Serbia and Bosnia- Herzegovina we applied depth-damage functions derived from data for other countries and averaged over all the European countries. Therefore, the availability of detailed, country-specific damage reports at building scale (i.e. reporting hazard variables and the consequent damage for different building categories) would allow to derive specific damage functions for these countries and improve damage estimates (see Section 4.4, lines 584-587).*

ff) P15 L433: You should also reflect on the completeness of official damage reports.

*We included a brief discussion on this point in page 16, lines 478-486 of the revised paper: "The observed underestimation has to be evaluated considering the limitations of both observed data and damage assessment methodology. On one hand, the damage functions available for Croatia are not specifically designed for the country, as discussed in Section 2.3.Also,estimated damages include only direct damage to buildings, while infrastructural damage is only partially accounted for (e.g. damage to the dyke system). On the other hand, official estimates are affected by the absence of clear standards for loss assessment and reporting (Corbane et al., 2015; IRDR, 2015).Thieken et al. (2016) observed that reported losses are rarely complete and that it may require years before reliable loss estimates are available for an event".*

gg) P15 L443: why? It would be interesting to see if the reference simulation is within the range of 25-75 quantiles.

*The revised paper now includes results from the simulations of 25 and 75 quantiles for May 13.*

hh) P17 L476: please state how many days

*In the revised paper we added a specific discussion on the performance regarding lead time in page 17, lines 503-507: "However, it has to be considered that peak flow timing was rather variable across the Sava river basin, due to its extent. While in the Kolubara river the highest discharges occurred on 14and 15 May, peak flows in other tributaries were reached later (between 14thand 16th for Bosna River, on 16thfor Drina, on 17thfor Sana River), and on the main branch of the Sava River the flood peaks occurred after 17 May".*

ii) P17 L491: It would be valuable to refer to the existing international frameworks on impact data collection, see also: Thieken, A. H., Bessel, T., Kienzler, S., Kreibich, H., Müller, M.,

Pisi, S. and Schröter, K.: The flood of June 2013 in Germany: how much do we know about its impacts?, Nat. Hazards Earth Syst. Sci., 16(6), 1519–1540, doi:10.5194/nhess-16-1519-2016, 2016.

*We thank the Reviewer for the suggestion, in the revised paper we elaborated on this point adding references to the suggested paper and to reports by IRDR (2015) and Corbane et al. (2015).*

 jj)  P17 L496: please name the benefits

*We further elaborated this paragraph which now reads as follows (page 22, lines 620-624):"As such, forecast-based risk assessment should be regarded as plausible risk scenarios that can provide valuable information for local, national and international authorities, complementing standard flood warnings. In particular, the explicit quantification of impacts opens the road to a more effective use of early warning information in emergency management, allowing to evaluate costs and benefits of response measures."*
*Please note also that in the revised paper the benefits provided by the risk forecasting procedure are now mentioned and discussed in other sections (see reply to points "3" and "d" for more details)*

*Additional references*

   Corbane, C., de Groeve, T., and Ehrlich, D.: Guidance for Recording and Sharing Disaster Damage and Loss Data – Towards the development of operational indicators to translate the Sendai Framework into action, Report, JRC95505, EUR 27192 EN, 2015.
   Coughlan de Perez, E. van Aalst, M. K. et al., Action-based flood forecasting for triggering humanitarian action, Hydrology and Earth System Sciences 20, 3549-3560, 2016.doi:10.5194/hess-20-3549-2016
   De Bruijn, K. M., Diermanse, F. L. M., Beckers, J. V. L., An advanced method for flood risk analysis in river deltas, appliedto societal flood fatality risk in the Netherlands .Nat. Hazards Earth Syst. Sci., 14, 2767–2781, 2014, doi:10.5194/nhess-14-2767-2014.
   ESRI map of World Cities, accessed on 06/03/2017 at http://www.arcgis.com/home/item.html?id=dfab3b294ab24961899b2a98e9e8cd3d.
   European Commission, Copernicus Land Monitoring Service, accessed on 02/02/2017 at http://land.copernicus.eu/pan-european/corine-land-cover
    Huizinga, J., de Moel, H., Szewczyk, W. (2017). Global flood damage functions. Methodology and the database with guidelines. EUR 28552 EN. doi: 10.2760/16510
   Tanoue, M., Hirabayashi, Y., Ikeuchi, H., 2016. Global-scale river flood vulnerability in the last 50 years. Scientific Reports, 6, 36021.

Vogt et al., A pan-European river and catchment database, JRC Reference Reports 2007,doi:0.2788/35907

**Reply to Reviewer 2**

The authors present a first attempt to develop a flood impact forecasting procedure that is fully integrated in a continental scale flood early warning system. They demonstrate this system by benchmarking various components against a flood events in May 2014in Bosnia-Herzegovina, Croatia and Serbia. The paper builds on two directions of several previous works of the various authors: (1) the EFAS system that has previously been used for forecasting peak flows; and (2) the impact assessment module that has been used in several past risk studies for current and future conditions. In my opinion, this is a laudable effort – the need for such studies has been clearly vocalized in many past papers, and in many scientific and policy-related fora. I greatly appreciate the effort undertaken not simply to present the framework, but to try to benchmark it for an actual event. Of course, 1 event remains a limited benchmarking, but I believe that the benchmarking has been carried out in a way much more thorough to past studies in large scale risk modelling. The novelty here is not in the models themselves, which have been developed in pervious papers, but bringing them together for impact forecasting. The paper is well written and clear, and provides enough level of detail on the already developed models, without too much repetition.

*We thank Reviewer 2 for his/her positive comments on our work.*

I believe that the paper therefore is an important first step forward in this direction, and therefore merits publication in NHESS, subject to the authors being able to address the following issues:

1) L119-121: "In case thresholds are exceeded persistently over several forecasts, flood warnings for the affected locations are issued to the members of the EFAS consortium." Please explain this statement better: which thresholds? And what is meant by "over several forecasts"?.

*To address this comment we rephrased part of the section which now reads as follows (page 4, lines 122-129):"The reference simulation is also used to estimate discharge values for the return periods corresponding to 1, 2, 5 and 20-year at every point of the river network. All flood forecasts are compared against these discharge thresholds and the threshold exceedance is calculated. In case the 5 year threshold is consistently exceeded over 3 consecutive forecasts, flood warnings for the affected locations are issued to the members of the EFAS consortium. The persistence criterion has been introduced to reduce the number of false alarms and focus*

*on large fluvial floods caused mainly by widespread severe precipitation, combined rainfall with snow-melting or prolonged rainfalls of medium intensity".*

2) L161-162: "We first identify the maximum discharge predicted over the full forecasting period, calculated using the median discharge from ensemble forecasts at each river grid cell". It is not clear to me from this sentence how this works. Do you take the maximum discharge across the entire ensemble for each lead time? (e.g. for lead time 1 day take the max discharge of all the ensemble members at 1 day lead) Or is something else meant here? Please clarify.

*This sentence has been rephrased as follows (page 6, lines 178-180: "At each grid cell, we first identify the median of the ensemble forecast given by the latest EFAS prediction, and then select the maximum discharge of the median over the full forecasting period (10 days)".*

3) It is stated that the flood protection standards of Jongman et al. (2014) are used, and integrated with information from literature review and local authorities where available. In terms of transparency and reproducibility, I recommend a list (e.g Supplementary Information or in Appendix) showing the regions in which the values from Jongman et al were replaced, and which values were used.

*Following the Reviewer's suggestion, the revised paper now includes an appendix with a list of the updates and additions to the flood protection level map developed by Jongman et al.*
*The list will show the regions where values have been updated, the old and new values, and the source of information.*

4) In the validation of the inundation maps, the authors have chosen only to report the hit rates. I find this problematic, as a (theoretical) model that greatly overestimates flood extent would tend to have very high hit rates. Therefore, in itself it only tells half the story. I believe that it would be more prudent to also report the false alarm ratios. This is especially important, since in Table 3 it is shown that the simulations show a much larger flooded area than the observed datasets, which could be leading to the high hit rates.

*We agree with the Reviewer on that presenting the results also in term of overestimation is necessary. To this end, in the revised version Table 3 now includes overestimation (or underestimation) ratios between simulations and all the available observations, to provide a more objective presentation of the results.*
*However, regarding the results in Table 4 we believe that it is more correct not to compute false hit ratio because, as discussed in the manuscript, we know that the available satellite flood maps underestimated the actual flood extent. As such, false alarm ratio scores would be low without being supported by reliable observations, giving an incorrect view of the performance (see page 12, lines 339-342).*

5) With regards the validation of the flood risk (I think it would be better called "flood impacts"), expressed as affected population, on lines 414-415 it is stated that: ". . .results from the reference simulation match well figures reported for all the flooded counties of Croatia except for the Vukovar-Srijem County." This is a very subjective statement: how is "match well" defined? For example, in the Osjek-Baranja Country, the observed dataset reports 200 people, whilst the simulated dataset suggests 1300 – i.e. a difference of 550%. I realise that the definitions used in the simulated/observed datasets are different, and so the direct comparison is difficult, but it would be more transparent to report the differences openly than disguise relatively large differences with ambiguous language.

*We agree on that the evaluation of results requires the use of a more precise language. In Section 4.2 of the revised manuscript we modified accordingly the presentation of results, commenting the limitations of simulated impacts and focusing on the areas with larger differences between simulations and observations (page 16, lines 459-470).*
*Also, we carefully revised the use of terms "flood risk" and "flood impact" in the paper (see also the reply to points 1 and A raised by Reviewer 1 for a more detailed discussion).*

6) One of the reasons given for the large difference in simulated damage between the reported and simulated dataset is that the damage curves applied have not yet been calibrated for Bosnia-Herzegovina, Croatia and Serbia. If this is the case, is it even useful to include this information in the warning?

*This comment has been addressed in the new Section 4.4, page 21 lines 575-587: "When designing the structure and output of risk assessment, it has to be considered that the type and amount of information provided must be based on users' requests. As a matter of fact, different end users may be interested in different facets of flood impact (Molinari et al., 2014), but at the same time it is important to avoid information overload during emergency management. Again, finding a compromise requires a close collaboration with the user community.*
*For instance, damage estimation has been included in the impact assessment upon request of EFAS end users, despite the known limitations of the damage functions dataset, in particular the absence of country-specific damage functions for the majority of countries in Europe. From this point of view, the case study described in this work is representative of the level of precision that may be achievable in these countries. Future improvements can be possible with the availability of detailed, country-specific damage reports at building scale (i.e. reporting hazard variables and the consequent damage for different building categories) would allow to derive specific damage functions."*

7) In the conclusion, it is stated that the "Comparison of reported and simulated flooded areas suggests that the methodology enables to identify areas at risk well in advance. . ." Whilst the results do indeed show some encouraging skill, I think the phrase "well in advance" seems like oversell. The 12th May forecast for the 14th May flood showed little sign of flooding. The impacts were rather clear on the 13th May, giving a good confidence warning 1 day in advance. It is of course subjective whether 1 day is "well in advance" – it depends on the actions that planners need to take.

*We apologize for not having been precise on presenting the performance regarding lead time. To solve this issue, in the revised manuscript we modified this part of the conclusion by reporting the lead times provided by EFAS forecasts without additional comments, and we added a dedicated description at page 17, lines 503-512:" However, it has to be considered that peak flow timing was rather variable across the Sava river basin, due to its extent. While in the Kolubara river the highest discharges occurred on 14and 15 May, peak flows in other tributaries were reached later (between 14thand 16th for Bosna River, on 16thfor Drina, on 17thfor Sana River), and on the main branch of the Sava River the flood peaks occurred after 17 May. Thus, in a hypothetical scenario where EFAS risk forecast were routinely used for emergency management, on one hand there would have been still time to improve risk estimates thanks to updated flood forecasts. On the other hand, the forecast released on 13 May would have given to emergency responders a warning time of at least 2 days to plan response measures in several affected areas, chiefly in the Kolubara and Bosna basins."*

Minor comments:

a) L60: the authors refer to a paper by Ward et al., 2016 to support the claim that "flood impact forecasts are increasingly being requested by end users of early warning systems". This facet is already discussed in Ward et al (2015), which would seem a more prudent paper to cite.

*We agree with the Reviewer, in the revised manuscript we replaced the reference as suggested.*

b) L131: "we decided create" to "we decided to create"; L222: wide spread to widespread; L368: "time o image" to "time of image".

*These typos have been corrected.*

c) L179: Batista e Silva et al. (2012)→Batista and Silva et al. (2012)

*The reference is actually correct, first author's surname is "Batista e Silva".*

**An operational procedure for rapid flood risk assessment in Europe**

**Francesco Dottori, Milan Kalas, Peter Salamon, Alessandra Bianchi,Lorenzo Alfieri ,Luc Feyen**

European Commission, Joint Research Centre, Directorate for Space, Security and Migration, Via E. Fermi 2749, 21027 Ispra, Italy.

francesco.dottori@ec.europa.eu

**Keywords:** real-time, early warning system, flood hazard mapping, flood impact, economic damage, population, risk assessment

**Abstract**

The development of methods for rapid flood mapping and risk assessment is a key step to increase the usefulness of flood early warning systems, and is crucial for effective emergency response and flood impact mitigation. Currently, flood early warning systems rarely include real–time components to assess potential impacts generated by forecast flood events. To overcome this limitation, this work describes the benchmarking of an operational procedure for rapid flood risk assessment based on predictions issued by the European Flood Awareness System (EFAS). Daily streamflow forecasts produced for major European river networks are translated into event-based flood hazard maps using a large map catalogue derived from high-resolution hydrodynamic simulations. Flood hazard maps are then combined with exposure and vulnerability information, and the impacts of the forecast flood events are evaluated in terms of flood prone areas, economic damage and affected population, infrastructures and cities.

An extensive testing of the operational procedure is carried out by analysing the catastrophic floods of May 2014 in Bosnia-Herzegovina, Croatia and Serbia. The reliability of the flood mapping methodology is tested against satellite-based and report-based flood extent data, while modelled estimates of economic damage and affected population are compared against ground-based estimations. Finally, we evaluate the skill of risk estimates derived from EFAS flood forecasts with different lead times and combinations of probabilistic forecasts. Results show the potential of the real-time operational procedure in helping emergency response and management.

> **Commented [FD1]:** R1-P1 addressed

**1) Introduction**

Nowadays, flood early warning systems (EWS) have become key components of flood management strategies in many rivers (Cloke et al., 2013; Alfieri et al., 2014a).They can increase preparedness of authorities and population, thus helping reduce negative impacts (Pappenberger et al., 2015). Early warning is particularly important for cross-border river basins where
cooperation between authorities of different countries may require more time to inform and
coordinate actions (Thielen et al., 2009).
In this context, the European Commission has developed the European Flood Awareness System
(EFAS) which provides operational flood predictions in major European rivers as part of the
Copernicus Emergency Management Services. The service is fully operational since 2012 and
available to hydro-meteorological services with responsibility in flood warning, EU civil
protection and their network.
While early warning systems are routinely used to predict flood magnitude, there is still a gap in
the ability to translate flood forecasts into risk forecasts, that is, to evaluate the possible
consequences generated by forecast events (e.g. flood prone areas, affected population, flood
damages losses), given their probability of occurrence. Generally, flood impacts are evaluated
considering reference risk scenarios where a fixed return period is used for all the area of interest,
for instance based on official maps issued by competent authorities (EC 2007). However, this
implies some degree of interpretation to define flood impact and risk in case of a flood forecast.
A few research projects are being developed where flood impact estimation is automated and
linked to event forecasting (Rossi et al., 2015; Schulz et al., 2015; Saint-Martin et al., 2016),
however to our knowledge these systems are still at experimental phase, and not yet integrated
into operational EWS.
The availability of real-time operational systems for assessing potential consequences of forecast
events would be a substantial advance in helping emergency response (Molinari et al., 2013), and
indeed flood risk forecasts are increasingly being requested by end users of early warning systems
(Emerton et al., 2016; Ward et al., 2015). At local scale, the joint evaluation of flood probabilities
and consequences may not only increase preparedness of emergency services, but also allow cost-
benefit considerations for planning and prioritizing response measures (e.g. strengthening flood
defences, planning evacuation of people at risk). At European scale, the possibility to receive
prior information on expected flood risk would help the Emergency Response Coordination
Centre (ERCC) in prioritizing and coordinating support to national emergency services.
In the present paper, we describe a methodology designed to meet the needs of EWS users and
overcome the limitations mentioned so far. The methodology translates EFAS flood forecasts into
event-based flood hazard maps, and combines hazard, exposure and vulnerability information to
produce risk estimations in near-real time. All the components are fully integrated within the
EFAS forecasting system, thus providing seamless risk forecasts at European scale.
To demonstrate the reliability of the proposed methodology, we perform a detailed assessment
focused on the 2014 floods in the Sava River Basin in Southeast Europe. A large dataset for the
evaluation of the results has been collected, which consists of observed flood magnitude, flood
extent derived from different satellite imagery datasets, and detailed post-event evaluation of
flood impacts, economic damage assessment and affected population and infrastructure.
The reliability of the flood mapping procedure is first assessed by assuming a "perfect" forecast,
where flood magnitude is taken from real observations instead of EFAS predictions. The effect

Commented [FD2]: R1-A: addressed

Commented [FD3]: R1-B addressed

Commented [FD4]: R1-C addressed

Commented [FD5]: R1-E addressed

[revised manuscript text omitted]

---

## Referee Report (RR1)

[referee-annotated manuscript omitted]

---

## Author Response (AR2)

**Author's response**

Please find below the response to the reviews, followed by the marked-up manuscript version.

**Reply to Reviewer 1**

The authors have done a convincing job in revising the paper. All major issues raised in the peer review have been thoroughly addressed. I spotted some typos in the manuscript which should be corrected. Two sentences in the conclusions need rephrasing. Please find the details in the annotated pdf.
*We report here below the corrections asked by Reviewer 1.*

1) lines 472-474: what is the usual range of uncertainty? I assume this is very context specific.

*We modified this sentence by referring to previous studies where the range of uncertainty was evaluated. The period now reads as follows" The difference is relevant but still within the range of uncertainty of damage models quantified in previous studies (de Moel and Aerts, 2011; Wagenaar et al., 2016)".*

2) lines 625-631: please check and rephrase.

*This part of the conclusion actually had several typos. The two sentences have been revised as follows: "For the immediate future, we plan to test a number of modifications and alternative approaches for the hazard mapping and risk assessment components. For instance, flood hazard maps are now computed using only the median of EFAS ensemble forecasts, but in principle the methodology can also be applied to more ensemble members, in order to take account of (for example) flood scenarios that are less probable but potentially more severe, and to provide a more complete risk evaluation (such as the application described this paper)".*

**Reply to Reviewer 2**

The authors have done a good job in addressing my concerns on the original manuscript, and I believe that this paper would be a valuable addition to the literature. I am happy to recommend publication, subject to the following minor points being addressed:

1.      Lines 568-573. Here, several measures that could be taken based on the forecasts are described: "For instance, we are currently evaluating the use of EFAS risk forecast to trigger satellite rapid flood mapping through Copernicus EMS, with the aim of improving response time and detection of flooded areas. More demanding measures (e.g. monitoring flood defences, deployment of emergency services in areas at risk planning evacuation of people at risk), could instead be put in place upon confirmation from local flood monitoring systems". The first part of this seems fine, but I don't see the value of the second part, or it needs to be explained more clearly. If a local flood monitoring system already exists for a specific region, then why are the EFAS information for that region needed? I would assume that EFAS is of use in those regions where there is no local flood monitoring system.

*The reasoning was not well explained. We rewrote the second part as follows: "In the future, especially in areas where no local monitoring systems are available, EFAS risk forecasts may be used to plan more demanding measures such as monitoring of flood defences, deployment of emergency services and evacuation of endangered people. Even where local systems are operating, risk forecasts may provide additional, valuable information with respect to standard streamflow forecasts. However, in these areas emergency measures should be enacted on confirmation from local monitoring systems."*

2.      Lines 590-593: "…whereas probabilistic risk methods (e.g. de Bruijn et al., 2014) and the use of mortality rates calculated from previous flood events (e.g. Tanoue et al., 2016)" Tanoue et al actually apply the method developed by Jongman et al. (2015), which should therefore be cited here.

*We agree with the Reviewer and we added the suggested reference.*

3.      Whilst well written, there are quite a lot of minor grammatical points; a thorough proof-reading by a native speaker would enhance the manuscript further.

*The English language of the manuscript has been improved as suggested.*

[revised manuscript text omitted]